# Transcription termination and antitermination of bacterial CRISPR arrays

Anne M Stringer[1], Gabriele Baniulyte[2], Erica Lasek-Nesselquist[1], Kimberley D Seed[3,4], Joseph T Wade[1,2]*

[1]Wadsworth Center, New York State Department of Health, Albany, United States; [2]Department of Biomedical Sciences, School of Public Health, University at Albany, Albany, United States; [3]Department of Plant and Microbial Biology, University of California, Berkeley, Berkeley, United States; [4]Chan Zuckerberg Biohub, San Francisco, United States

**Abstract** A hallmark of CRISPR-Cas immunity systems is the CRISPR array, a genomic locus consisting of short, repeated sequences ('repeats') interspersed with short, variable sequences ('spacers'). CRISPR arrays are transcribed and processed into individual CRISPR RNAs that each include a single spacer, and direct Cas proteins to complementary sequences in invading nucleic acid. Most bacterial CRISPR array transcripts are unusually long for untranslated RNA, suggesting the existence of mechanisms to prevent premature transcription termination by Rho, a conserved bacterial transcription termination factor that rapidly terminates untranslated RNA. We show that Rho can prematurely terminate transcription of bacterial CRISPR arrays, and we identify a widespread antitermination mechanism that antagonizes Rho to facilitate complete transcription of CRISPR arrays. Thus, our data highlight the importance of transcription termination and antitermination in the evolution of bacterial CRISPR-Cas systems.

*For correspondence:
joseph.wade@health.ny.gov

**Competing interests:** The authors declare that no competing interests exist.

## Introduction

CRISPR-Cas systems are adaptive immune systems found in many bacteria and archaea (*Wright et al., 2016*). The hallmark of CRISPR-Cas systems is the CRISPR array, which is composed of multiple alternating, short, identical 'repeat' sequences, interspersed with short, variable 'spacer' sequences. A critical step in CRISPR immunity is biogenesis (*Wright et al., 2016*), which involves transcription of a CRISPR array into a single, long precursor RNA that is then processed into individual CRISPR RNAs (crRNAs), with each crRNA containing a single spacer sequence. crRNAs associate with an effector Cas protein or Cas protein complex, and direct the Cas protein(s) to an invading nucleic acid sequence that is complementary to the crRNA spacer and often includes a neighboring Protospacer Adjacent Motif (PAM). This leads to cleavage of the invading nucleic acid by a Cas protein nuclease, in a process known as 'interference'.

CRISPR arrays can be expanded by acquisition of new spacer/repeat elements at one end of the array, in a process known as adaptation. The ability to become immune to newly encountered invaders is presumably a strong selective pressure that promotes adaptation (*Bradde et al., 2020*; *Martynov et al., 2017*), although shorter CRISPR arrays appear to be strongly favored in bacteria (*Weissman et al., 2018*). Little is known about factors that negatively influence CRISPR array length. It has been hypothesized that increased array length is selected against because of the potential for individual crRNAs to become less effective as the effector complex is diluted among more crRNA variants (*Bradde et al., 2020*; *Martynov et al., 2017*; *Rao et al., 2017*). The diversity of potential invaders and the mutation frequency of invaders have also been proposed to impose selective pressure on array length (*Martynov et al., 2017*). Lastly, array length is known to be impacted by

deletions caused by homologous recombination between repeats (*Gudbergsdottir et al., 2011*; *Kupczok et al., 2015*).

Rho is a broadly conserved bacterial transcription termination factor. Rho terminates transcription only when nascent RNA is untranslated (*Mitra et al., 2017*). Hence, the primary function of Rho is to suppress the transcription of spurious, non-coding RNAs that initiate as a result of pervasive transcription (*Lybecker et al., 2014*; *Peters et al., 2012*; *Wade and Grainger, 2014*). To terminate transcription, Rho must load onto nascent RNA at a 'Rho utilization site' (Rut). The precise sequence and structure requirements for Rho loading are not fully understood, but Ruts typically have a high C:G ratio, limited secondary structure, and are enriched in YC dinucleotides (*Mitra et al., 2017*; *Nadiras et al., 2018*). However, the overall sequence/structure specificity of Ruts is believed to be low, and a large proportion of the *Salmonella* Typhimurium genome is predicted to be capable of functioning as a Rut (*Nadiras et al., 2018*). Once Rho loads onto nascent RNA, it translocates along the RNA in a 5' to 3' direction using its helicase activity. Rho typically catches the RNA polymerase (RNAP) within 60–90 nucleotides, leading to transcription termination, with termination typically occurring at an RNAP pause site (*Mitra et al., 2017*).

The activity of Rho can be inhibited by a variety of mechanisms that collectively are referred to as 'antitermination'. Antitermination mechanisms can be grouped into two classes: targeted and processive (*Goodson and Winkler, 2018*). Targeted antitermination affects a single site and does not otherwise alter the properties of the transcription complex. For example, targeted antitermination could involve occlusion of a single Rho loading site by an RNA-binding protein. Processive antitermination, on the other hand, involves modification of the transcription machinery such that RNAP becomes resistant to termination for the remainder of that transcription cycle (*Goodson and Winkler, 2018*). This typically occurs due to association of a protein or protein complex with the elongating RNAP due to sequence-specific *cis*-acting elements in the DNA or nascent RNA. One of the best-studied processive antitermination mechanisms occurs on ribosomal RNA (rRNA) and involves the Nus factor complex. The Nus factor complex consists of five proteins, NusA, NusB, NusE (ribosomal protein S10), NusG, and SuhB, that bind to both nascent RNA and elongating RNAP. Nus complex formation begins with sequence-specific association of NusB/E with a short RNA element known as 'BoxA'. Association of the Nus complex with both RNAP and the BoxA leads to formation of a loop in the nascent RNA (*Bubunenko et al., 2013*; *Burmann et al., 2010*; *Huang et al., 2020*; *Singh et al., 2016*). The most recently identified member of the Nus complex, SuhB, is recruited to elongating RNAP in a *boxA*-dependent manner (*Singh et al., 2016*), interacts with NusA, NusG, RNAP, and the nascent RNA, and is required for assembly and activity of the Nus factor complex (*Dudenhoeffer et al., 2019*; *Huang et al., 2020*; *Huang et al., 2019*; *Wang et al., 2007*). The Nus factor complex prevents Rho termination (*Squires et al., 1993*; *Torres et al., 2004*) in a BoxA-dependent manner (*Aksoy et al., 1984*; *Li et al., 1984*; *Squires et al., 1993*), and BoxA elements are found in phylogenetically diverse copies of rRNA (*Arnvig et al., 2008*; *Sen et al., 2008*).

Like rRNA, CRISPR array transcripts are non-coding and often long, making them ideal substrates for Rho (*Pougach et al., 2010*). Here, we show that Rho termination provides selective pressure against increased CRISPR array length in bacteria. Moreover, we show that BoxA-mediated antitermination is a widespread mechanism by which bacteria protect their CRISPR arrays from Rho termination. Disrupting BoxA-mediated antitermination leads to premature Rho termination within CRISPR arrays that renders later spacers ineffective for CRISPR immunity.

## Results

### *Salmonella* Typhimurium CRISPR arrays have functional *boxA* sequences upstream

We identified *boxA*-like sequences a short distance upstream of both CRISPR arrays (CRISPR-I and CRISPR-II) in *S.* Typhimurium, which encodes a single, type I-E CRISPR-Cas system (*Figure 1A*). The putative *boxA* sequences are 78 and 77 bp upstream of the first repeat in the CRISPR-I and CRISPR-II arrays, respectively. To facilitate studies of the *S.* Typhimurium CRISPR-Cas system, which is transcriptionally silenced by H-NS (*Navarre et al., 2006*), we introduced a strong, constitutive promoter (*Luo et al., 2015*) in place of *cas3* (*Figure 1A*). This promoter drives transcription of the *cas8e-cse2-cas7-cas5-cas6e-cas1-cas2* operon, and our ChIP-qPCR data for RNAP indicate that transcription

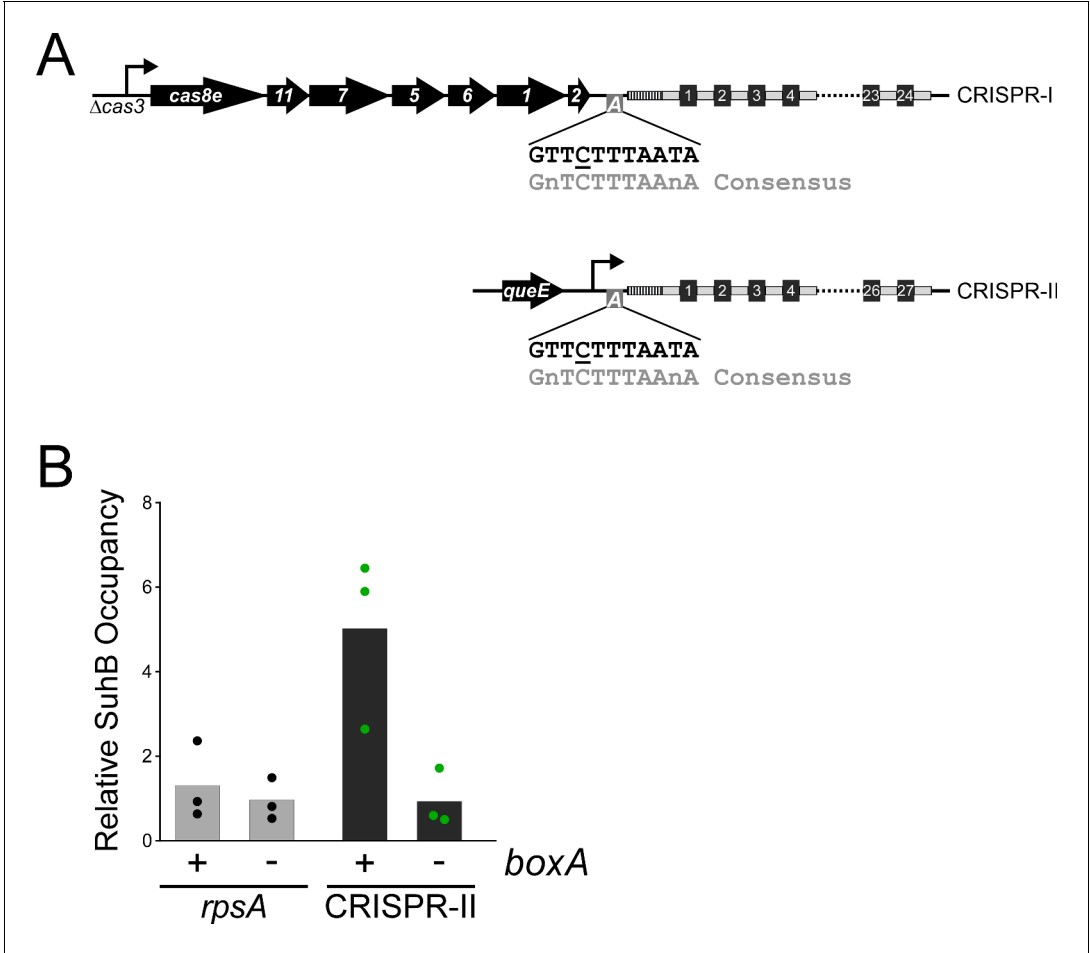

**Figure 1.** *boxA* elements upstream of both CRISPR arrays in *Salmonella* Typhimurium. (**A**) Schematic of the two CRISPR arrays in *S.* Typhimurium. Repeat sequences are represented by gray rectangles and spacer sequences are represented by black squares. Spacers are numbered within the array, with spacer one being closest to the leader sequence (dashed rectangle). The CRISPR-I array is co-transcribed with the upstream *cas* genes. For the work presented in this study, the *cas3* gene was deleted and replaced with a constitutive promoter. Similarly, a constitutive promoter was inserted immediately downstream of *queE*, upstream of the CRISPR-II array. *boxA* elements (boxes containing an 'A') are located immediately upstream of the leader sequences of both CRISPR arrays. (**B**) Relative occupancy of SuhB-TAP, determined by ChIP-qPCR, within the highly expressed *rpsA* gene (gray bars), or at the *boxA* sequence upstream of CRISPR-II (black bars). Occupancy was measured in a strain with an intact *boxA* upstream of CRISPR-II (AMD710), or a single base-pair substitution within the *boxA* (underlined base in (A); AMD711). Values shown with bars are the average of three independent biological replicates, with dots showing each individual datapoint.

The online version of this article includes the following figure supplement(s) for figure 1:

**Figure supplement 1.** The CRISPR-I array is co-transcribed with the upstream *cas* gene operon.

continues through the *boxA* into the CRISPR array (*Figure 1—figure supplement 1*); in wild-type *S.* Typhimurium cells that lack the constitutive promoter upstream of *cas8e*, we detected little RNAP occupancy within the CRISPR array, suggesting that there is no active promoter between *cas2* and the start of the array. We also introduced a strong, constitutive promoter upstream of the CRISPR-II array, immediately downstream of the *queE* gene (*Figure 1A*), reasoning that this would mimic transcriptional readthrough from *queE*. Transcription from this promoter also covers the putative *boxA*. To determine whether the putative *boxA* elements upstream of the CRISPR arrays are genuine, we measured association of TAP-tagged SuhB with elongating RNAP at the CRISPR-II array using ChIP-qPCR, which detects indirect association of SuhB with the DNA (*Singh et al., 2016*). Our data indicate robust association of SuhB with the region immediately downstream of the putative *boxA*, but not with the highly transcribed *rpsA* gene that is not associated with a *boxA* (*Figure 1B*). By contrast, we detected substantially reduced SuhB association with the same genomic region in a strain

containing a single base pair substitution in the *boxA* that is expected to abrogate NusB/E association (*Baniulyte et al., 2017*; *Berg et al., 1989*; *Nodwell and Greenblatt, 1993*), with SuhB association being similar to that at *rpsA*. The level of SuhB association with *rpsA* was not substantially altered by the mutation in the CRISPR-II *boxA*. We conclude that the CRISPR-II array transcript includes a functional upstream BoxA. For almost 40 years, the Nus factor complex was believed to be a dedicated rRNA regulator, with no other known bacterial targets (*Sen et al., 2008*). We recently identified a novel function for the Nus factor complex – autoregulation of *suhB* – and we provided evidence for many additional targets (*Baniulyte et al., 2017*). Identification of CRISPR arrays as a novel target for the Nus factor complex further increases the number of known targets and provides new opportunities for investigating the mechanism by which Nus factors prevent Rho termination.

## BoxA-mediated antitermination of *S.* Typhimurium CRISPR arrays

We hypothesized that BoxA-mediated association of the Nus factor complex with RNAP at the *S.* Typhimurium CRISPR arrays prevents premature Rho-dependent transcription termination. To test this hypothesis, we constructed *lacZ* transcriptional reporter fusions that contain a constitutive promoter followed by the sequence downstream of the *queE* gene (upstream of the CRISPR-II array), extending to either the 2nd ('short fusion') or the 11th ('long fusion') spacer of the array (*Figure 2A*). We constructed equivalent fusions that contain a single base pair substitution in the *boxA* that is expected to abrogate NusB/E association (*Figure 1B*; *Baniulyte et al., 2017*; *Berg et al., 1989*; *Nodwell and Greenblatt, 1993*). We then measured β-galactosidase activity for each of the four fusions in cells grown with/without bicyclomycin (BCM), a specific inhibitor of Rho (*Mitra et al., 2017*). In the absence of BCM, expression of the long fusion but not the short fusion was substantially reduced by mutation of the *boxA* (*Figure 2B*). By contrast, expression of all fusions was similar for cells grown in the presence of BCM (*Figure 2B*). Thus, our data are consistent with BoxA-mediated, Nus factor antitermination of the CRISPR array, with Rho termination occurring between the 2nd and 11th spacer when antitermination is disrupted. Surprisingly, expression levels of both the short and long fusions were substantially higher in cells grown with BCM, even with an intact *boxA* (*Figure 2B*). By contrast, expression of a control reporter fusion that includes an intrinsic terminator upstream of *lacZ* was not affected by BCM treatment (*Figure 2C*); the intrinsic terminator substantially reduces, but does not abolish, *lacZ* expression (*Stringer et al., 2014*). We conclude that some Rho termination occurs upstream of the *boxA* in both the long and short CRISPR array reporter fusions. We also observed that expression of the long fusion was lower than that of the short fusion, even with an intact *boxA*, suggesting that Nus factors are unable to prevent all instances of Rho termination within the CRISPR array.

## Antitermination of *S.* Typhimurium CRISPR arrays facilitates the use of spacers throughout the arrays

Our data indicate that CRISPR arrays in *S.* Typhimurium are protected from premature Rho termination by Nus factor association with RNAP via the BoxA sequences. However, this does not necessarily mean that CRISPR-Cas function is affected by antitermination, since low levels of crRNA may be sufficient for Cas proteins to bind target DNA. We previously investigated the specificity of the Cascade complex of Cas proteins in *Escherichia coli*. Our data indicated that Cascade can bind to DNA targets with as few as 5 bp between the crRNA and the target DNA; consequently, the endogenous *E. coli* crRNAs direct Cascade binding to >100 chromosomal sites (*Cooper et al., 2018*), with all of the binding events involving very limited base-pairing that is insufficient for DNA cleavage by the Cas3 nuclease. The *E. coli* CRISPR-Cas system is a type I-E system, similar to that in *S.* Typhimurium. Given this similarity, we hypothesized that *S.* Typhimurium Cascade would also bind chromosomal targets, and that we could measure the effectiveness of different CRISPR array spacers by measuring the degree to which those spacers direct Cascade binding to cognate chromosomal sites. We used ChIP-seq to measure FLAG$_3$-Cas5 (a Cascade subunit) association with the *S.* Typhimurium chromosome in cells expressing crRNAs from both CRISPR arrays. Note that we used a different constitutive promoter upstream of the CRISPR-II array to that described above for ChIP of SuhB, although the promoter was inserted at the same location (see Methods). As expected, we detected association of Cas5 with a large number (236) of chromosomal regions (*Figure 3A*, *Supplementary file 1*). These

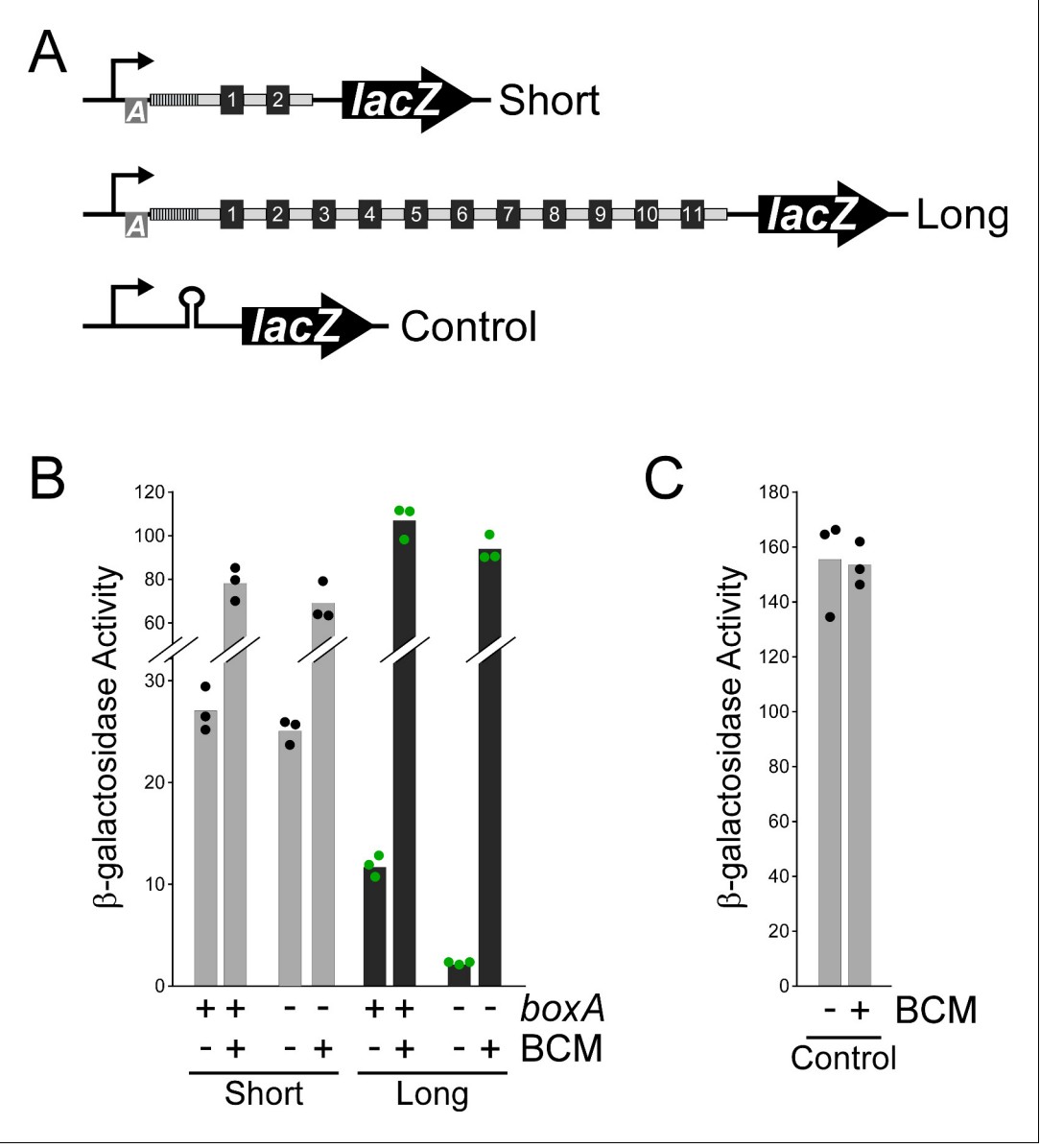

**Figure 2.** BoxA-mediated antitermination of a CRISPR array. (**A**) Schematic of short (pGB231 and pGB237) and long (pGB250 and pGB256) *lacZ* reporter gene transcriptional fusions to the CRISPR-II array, and a control transcriptional fusion (pJTW060) that includes an intrinsic terminator upstream of *lacZ*. (**B**) β-galactosidase activity of the short and long *lacZ* reporter gene fusions with either an intact (pGB231 and pGB250) or mutated (pGB237 and pGB256) *boxA* sequence, in cells grown with/without addition of the Rho inhibitor bicyclomycin (BCM). (**C**) β-galactosidase activity of the control *lacZ* reporter gene fusion (pJTW060) that includes an intrinsic terminator upstream of *lacZ*, in cells grown with/without BCM. Values shown with bars are the average of three independent biological replicates, with dots showing each individual datapoint.

binding sites are associated with five strongly enriched sequence motifs that correspond to a PAM sequence and the seed regions of spacers 1, 2, 3, 4, and 11 from the CRISPR-I array (*Figure 3B*). These enriched motifs indicate that the optimal PAM in *S.* Typhimurium is AWG. We then attempted to identify the corresponding crRNA spacer for all 236 Cascade-binding sites (see Materials and methods). Thus, we were able to uniquely associate 152 binding sites with a single spacer from one of the two CRISPR arrays (*Supplementary file 2*). To test the quality of these spacer assignments, we repeated the ChIP-seq experiment in a strain where the CRISPR-I array had been deleted. Consistent with our spacer assignments, Cas5 association decreased specifically at all DNA

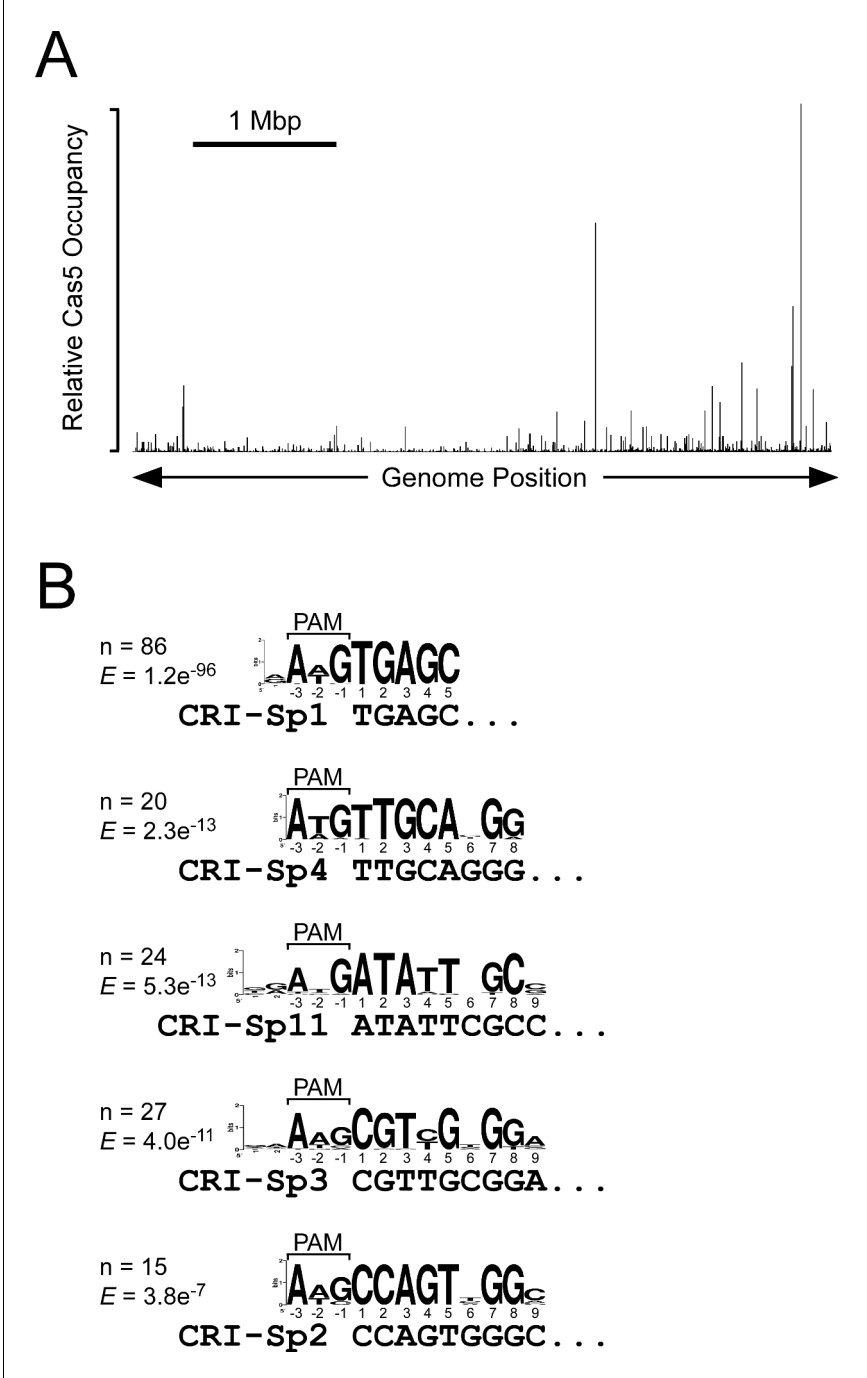

**Figure 3.** Extensive off-target binding of *S. Typhimurium* Cascade. (**A**) Relative FLAG$_3$-Cas5 occupancy across the *S.* Typhimurium genome as measured by ChIP-seq in strain AMD678. (**B**) Enriched sequence motifs in Cascade-bound regions, as identified by MEME (*Bailey and Elkan, 1994*). Sequence matches to the AWG PAM and to the seed sequence of specific spacers are indicated. The number of Cascade-bound genomic regions containing each sequence motif is indicated, as is the enrichment score ("*E*") generated by MEME (*Bailey and Elkan, 1994*). The online version of this article includes the following figure supplement(s) for figure 3:

**Figure supplement 1.** Confirmation of spacer assignments to Cas5- binding sites.

sites assigned to spacers from the CRISPR-I array (*Figure 3—figure supplement 1*). By comparing the ChIP-seq data from wild-type and CRISPR-I deleted cells, we were able to unambiguously associate an additional 32 Cascade-binding sites with a single spacer from one of the two CRISPR arrays (these binding sites had previously been associated with multiple possible spacers; *Supplementary file 2*).

We next measured FLAG$_3$-Cas5 association with the *S*. Typhimurium chromosome in cells containing a single base pair substitution in the *boxA* upstream of CRISPR-II that is expected to abrogate NusB/E association (*Figure 1B*; *Baniulyte et al., 2017*; *Berg et al., 1989*; *Nodwell and Greenblatt, 1993*). We observed no difference in Cas5 binding between the wild-type and mutant cells for sites associated with spacers from the CRISPR-I array, or sites associated with spacers 1–2 from the CRISPR-II array (*Figure 4A*; *Supplementary file 2*). By contrast, mutation of the CRISPR-II *boxA* led to decreases of between 3.7- and 10.7-fold in Cas5 binding to sites associated with spacers 9–23 from the CRISPR-II array (*Figure 4A–B*; *Supplementary file 2*). This effect was reversed by addition of BCM to the *boxA* mutant cells (*Figure 4C*; *Supplementary file 2*), indicating that reduced Cascade binding associated with spacers 9–23 of CRISPR-II is due to premature Rho termination of the array. Consistent with our reporter gene fusion data (*Figure 2B*), addition of BCM to cells with an intact *boxA* led to an increase in Cascade binding associated with spacers 9–23 of the CRISPR-II array (*Figure 4—figure supplement 1*; *Supplementary file 2*), supporting the notion that the BoxA prevents only a subset of premature Rho termination events.

We also measured FLAG$_3$-Cas5 association with the *S*. Typhimurium chromosome in cells containing a single base-pair substitution in the *boxA* upstream of the CRISPR-I array. Mutation of the CRISPR-I *boxA* had no impact on Cas5 binding to sites associated with spacers from the CRISPR-II array or spacers 1–5 from the CRISPR-I array (*Figure 4—figure supplement 2A*; *Supplementary file 2*). By contrast, mutation of the CRISPR-I *boxA* led to a decrease in Cas5 binding to sites associated with spacers 9–17 from the CRISPR-I array (*Figure 4—figure supplement 2A*), with the magnitude of the effect increasing as a function of the position of the spacer in the array (*Figure 4—figure supplement 2B*). This effect was reversed by addition of BCM to the *boxA* mutant cells (*Figure 4—figure supplement 2C*; *Supplementary file 2*), indicating that reduced Cascade binding using spacers 9–17 of CRISPR-I is due to premature Rho termination of the array. Unexpectedly, Cas5 binding associated with CRISPR-I spacers 18–23 was unaffected by the *boxA* mutation (*Figure 4—figure supplement 2B*), suggesting the existence of an additional promoter within spacer 16 or 17. Overall, the effect of mutating the CRISPR-I *boxA* was smaller than that of mutating the CRISPR-II *boxA*, suggesting that more RNAP can evade Rho termination at CRISPR-I.

To confirm that the effects of mutating the *boxA* sequences are due to the inability to recruit the Nus factor complex, we measured FLAG$_3$-Cas5 association with the *S*. Typhimurium chromosome in cells containing a defective Nus factor. We were unable to delete either *nusB* or *suhB*, suggesting that all Nus factors are essential in *S*. Typhimurium. Hence, we introduced a single-base substitution into the chromosomal copy of the *nusE* gene, resulting in the N3H amino acid substitution. The equivalent change in the *E. coli* NusE leads to a defect in Nus factor complex function (*Baniulyte et al., 2017*). Note that mutation of *nusE* also resulted in a ~ 147 kb deletion of the chromosome at an unlinked site (See Materials and methods for details of the deletion). Mutation of *nusE* led to a decrease in Cascade binding to sites associated with spacers 9–23 from the CRISPR-II array and spacers 13–17 from the CRISPR-I array (*Figure 5A*; *Supplementary file 2*). The effect of the *nusE* mutation on Cascade binding was smaller than that of the *boxA* mutations; however, the Nus factor complex is likely to retain partial function in the *nusE* mutant strain. To rule out the possibility that the chromosomal deletion in the *nusE* mutant strain was responsible for the effect on Cascade binding, we complemented the strain with plasmid-expressed wild-type NusE. Complementation led to an increase in the binding of Cas5 to sites associated with spacers 9–23 from the CRISPR-II array and spacers 13–17 from the CRISPR-I array (*Figure 5B*; *Supplementary file 2*), consistent with a specific effect of mutating *nusE*. Similarly, addition of BCM to the *nusE* mutant cells resulted in an increase in the binding of Cas5 to sites associated with spacers 9–23 from the CRISPR-II array and spacers 13–17 from the CRISPR-I array (*Figure 5C*; *Supplementary file 2*), indicating that reduced Cascade binding using spacers 9–23 of CRISPR-II and spacers 13–17 from the CRISPR-I array in the *nusE* mutant is due to premature Rho termination of the arrays. Based on the effects of mutating the *boxA* sequences and the effect of mutating *nusE*, we conclude that Nus factors prevent premature transcription termination of both *S*. Typhimurium CRISPR arrays, and that

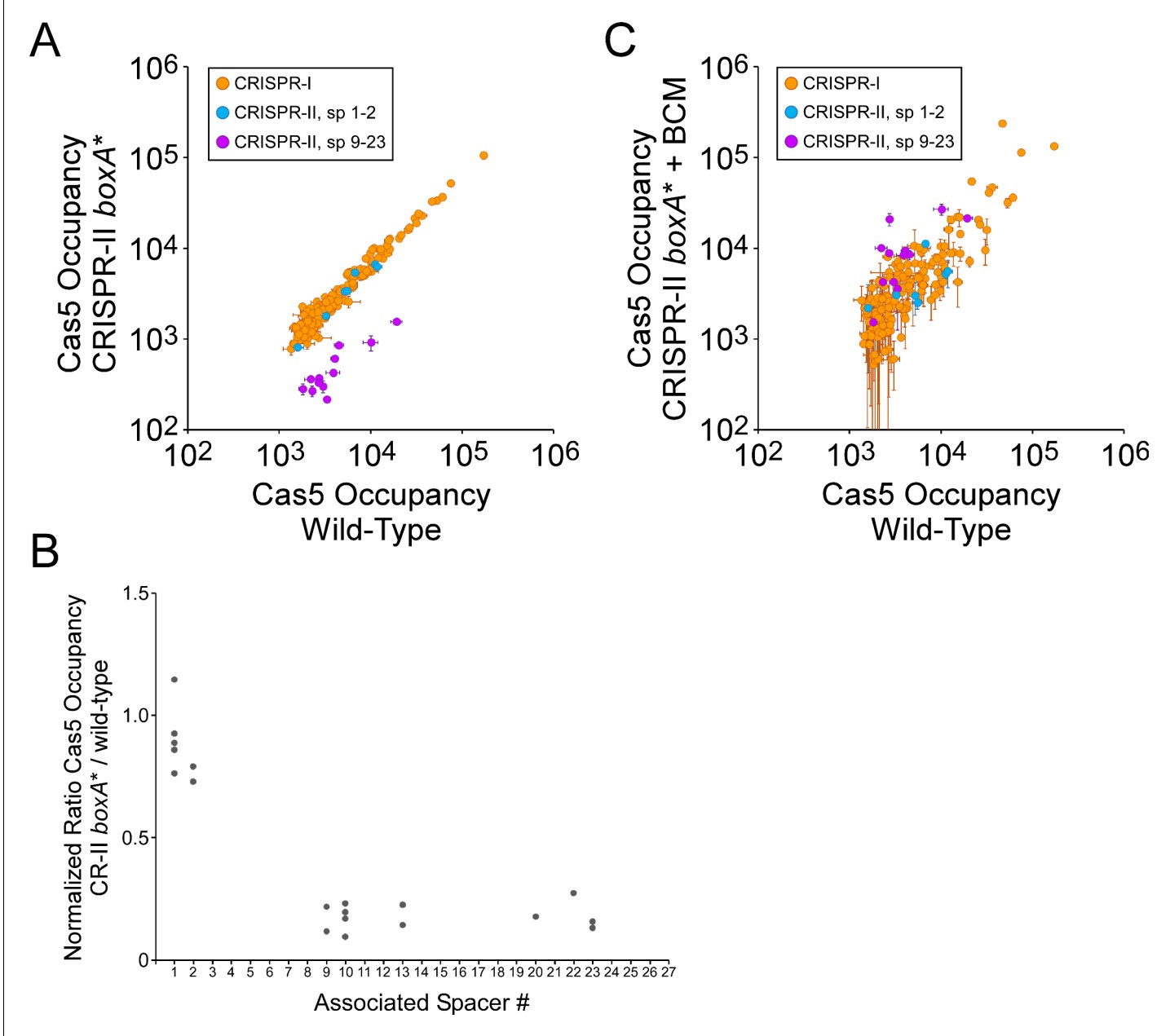

**Figure 4.** The CRISPR-II *boxA* facilitates use of all spacers by preventing premature Rho termination. (**A**) Comparison of FLAG₃-Cas5 ChIP-seq occupancy at off-target chromosomal sites in cells with an intact CRISPR-II *boxA* (AMD678; *x*-axis), and cells with a single base-pair substitution in the CRISPR-II *boxA* ('*boxA*\*'; AMD685; *y*-axis). Cascade binding associated with spacers from CRISPR-I is indicated by orange datapoints. Cascade binding associated with spacers 1–2 from CRISPR-II is indicated by light blue datapoints. Cascade binding associated with spacers 9–23 from CRISPR-II is indicated by purple datapoints. Values plotted are the average of two independent biological replicates. Error bars represent one standard-deviation from the mean. (**B**) Normalized ratio of FLAG₃-Cas5 occupancy associated with spacers from CRISPR-II. Values are plotted according to the associated spacer, and are normalized to the average value for sites associated with spacers from CRISPR-I. (**C**) Comparison of FLAG₃-Cas5 ChIP-seq occupancy at off-target chromosomal sites in cells with an intact CRISPR-II *boxA* (AMD678; *x*-axis), and cells with a single base-pair substitution in the CRISPR-II *boxA* ('*boxA*\*'; AMD685) that were treated with bicyclomycin (BCM; *y*-axis).

The online version of this article includes the following figure supplement(s) for figure 4:

**Figure supplement 1.** Inhibition of Rho in cells with an intact *boxA* increases the use of spacers 9–23 of CRISPR-II.

**Figure supplement 2.** The CRISPR-I *boxA* facilitates use of all spacers by preventing premature Rho termination.

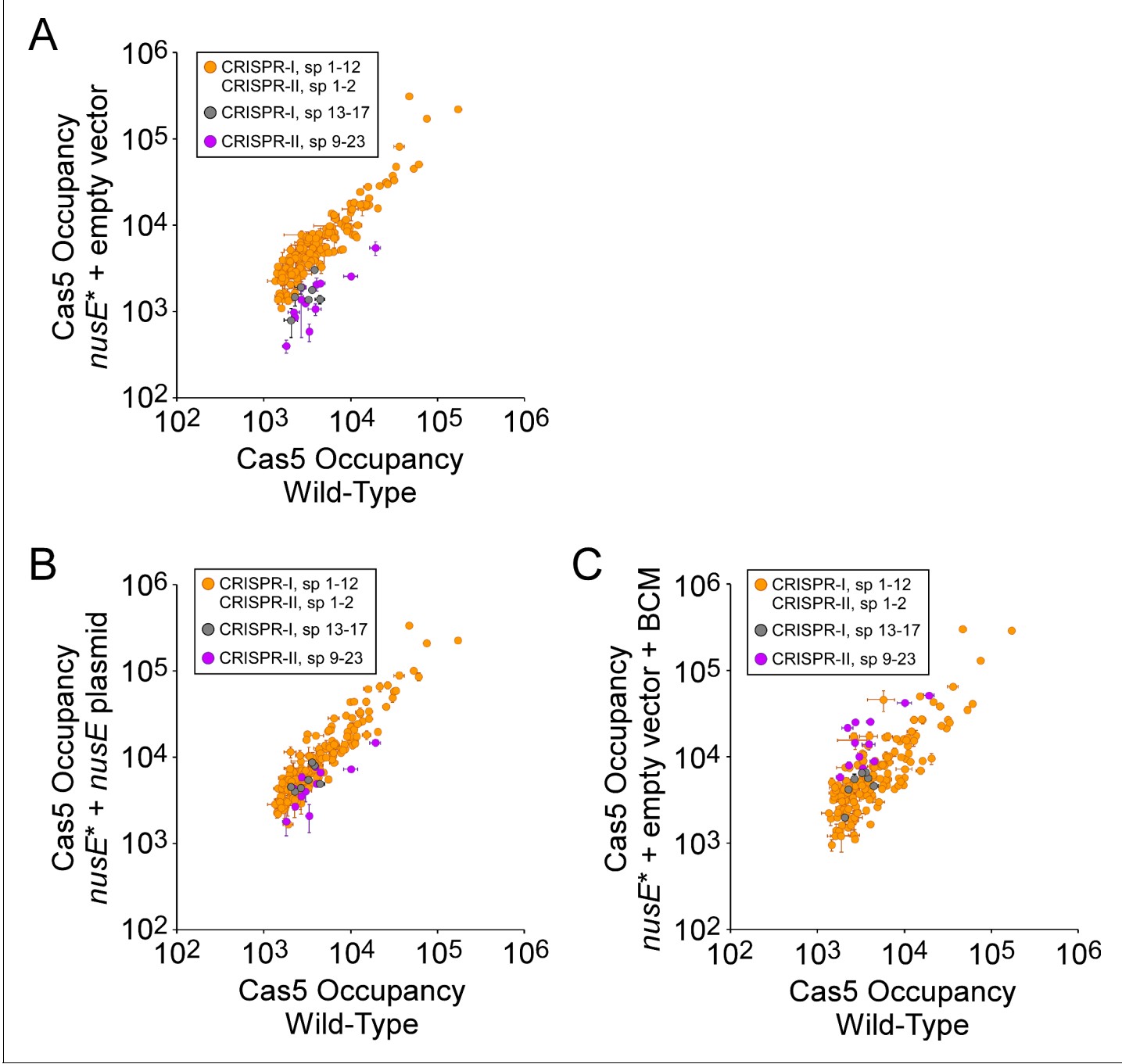

**Figure 5.** Nus factors facilitate use of all spacers by preventing premature Rho termination. (**A**) Comparison of FLAG$_3$-Cas5 ChIP-seq occupancy at off-target chromosomal sites in *nusE*$^+$ (AMD678; *x*-axis), and *nusE* mutant cells ('*nusE*\*'; AMD698) containing an empty vector (pBAD24-amp; *y*-axis). Cascade binding associated with spacers 1–12 from CRISPR-I, and spacers 1–2 from CRISPR-II, is indicated by orange datapoints. Cascade binding associated with spacers 13–17 from CRISPR-I is indicated by gray datapoints. Cascade binding associated with spacers 9–23 from CRISPR-II is indicated by purple datapoints. Values plotted are the average of two independent biological replicates. Error bars represent one standard-deviation from the mean. (**B**) Comparison of FLAG$_3$-Cas5 ChIP-seq occupancy at off-target chromosomal sites in *nusE*$^+$ (AMD678; *x*-axis) or *nusE* mutant cells ('*nusE*\*'; AMD698) expressing a plasmid-encoded (pAMD239) copy of wild-type *nusE* (*y*-axis). (**C**) Comparison of FLAG$_3$-Cas5 ChIP-seq occupancy at off-target chromosomal sites in *nusE*$^+$ cells (AMD678; *x*-axis), and *nusE* mutant cells ('*nusE*\*'; AMD698) containing an empty vector and treated with bicyclomycin (BCM; *y*-axis).

this has a direct impact on the ability of *S*. Typhimurium to use the majority of spacers in the CRISPR arrays.

## Immune activity of the *Vibrio cholerae* CRISPR-Cas system requires BoxA-mediated antitermination

Our data clearly indicate that BoxA-mediated antitermination is required for the activity of later spacers in the *S*. Typhimurium CRISPR arrays. However, we measured activity of spacers by their ability to direct Cascade binding, not interference. Moreover, we artificially induced expression of the *S*. Typhimurium *cas* genes and CRISPR arrays because their transcription is normally silenced by H-NS (*Lucchini et al., 2006*; *Navarre et al., 2006*); indeed, there is no evidence from phylogenetic analysis for recent activity of *Salmonella* CRISPR-Cas systems (*Shariat et al., 2015*). To address these limitations, we turned to the type I-E CRISPR-Cas system found in *V. cholerae* of the classical biotype. This CRISPR-Cas system is naturally active in immunity (*Box et al., 2016*), and has a putative *boxA* upstream of a 39-spacer CRISPR array (*Figure 6—figure supplement 1*). We constructed three derivatives of *V. cholerae* strain A50: (i) a Δ*cascade* mutant in which all *cas* genes are deleted except *cas1* and *cas2*, (ii) a Δ*cas1* mutant that lacks *cas1* but retains all other *cas* genes and hence is proficient for interference, and (iii) a *boxA*$^{mut}$ Δ*cas1* double mutant that has a two base pair substitution in the *boxA*. We constructed a collection of plasmids containing protospacers matching the 39 spacers in the *V. cholerae* A50 CRISPR array, each flanked by an optimal PAM (*Box et al., 2016*). We also constructed a control plasmid that lacks a protospacer. We pooled these plasmids and attempted to introduce them into each of the three strains by conjugation. Successful transconjugants were pooled, and the protospacer sequences were PCR-amplified and sequenced. We calculated the conjugation efficiency for each protospacer-containing plasmid into the Δ*cas1* (*boxA*$^+$) and *boxA*$^{mut}$ Δ*cas1* strains, normalizing to the conjugation efficiency of each plasmid into the Δ*cascade* strain and to the conjugation efficiency of the control plasmid that lacks a protospacer. We observed low conjugation efficiencies (0.01–10% of the control plasmid conjugation efficiency) for the majority of protospacers tested with the Δ*cas1* strain, with only protospacer 39, and to a lesser extent protospacer 38, avoiding CRISPR-Cas immunity (*Figure 6*). Thus, spacers 1–37 are active in CRISPR-Cas immunity in *V. cholerae*. For the *boxA*$^{mut}$ Δ*cas1* strain, conjugation efficiencies for protospacers 1–19 were essentially unchanged with respect to the *boxA*$^+$ strain. By contrast, conjugation efficiencies for spacers 20–37 were substantially higher (44.5-fold to 262-fold) in the *boxA*$^{mut}$ Δ*cas1* strain than in the *boxA*$^+$ strain (*Figure 6*). For spacers 28–36 and 38–39, conjugation efficiencies in the *boxA*$^{mut}$ Δ*cas1* strain were similar to those of the control plasmid, indicating a lack of interference (*Figure 6*; note that spacer 26 is an exact copy of spacer 10; *Figure 6—figure supplement 1*). We conclude that an intact *boxA* is needed to prevent premature Rho termination within the *V. cholerae* CRISPR array; in the absence of a functional BoxA, many of the spacers in the CRISPR array are rendered obsolete.

## BoxA-mediated antitermination of bacterial CRISPR arrays is phylogenetically widespread

Given that Rho is found in >90% of bacterial species (*D'Heygère et al., 2013*), and Nus factors are broadly conserved (*Figure 7—figure supplement 1*; *Supplementary file 3*; note that we assessed conservation of NusB rather than other members of the Nus factor complex because each of the other proteins has a second function, separate to its role in the complex), we reasoned that species other than *S. enterica* and *V. cholerae* may use the Nus factor complex to facilitate expression of their CRISPR arrays. Hence, we performed a phylogenetic analysis of sequences associated with CRISPR arrays. Among sequences found between a *cas2* gene and a downstream CRISPR array in 187 bacterial genera (each genus represented only once; see Materials and methods for details of sequence selection), there was a strongly enriched sequence motif that is a striking match to the known *boxA* consensus from *E. coli* (*Figure 7A*; *Supplementary file 4*; *Arnvig et al., 2008*; *Baniulyte et al., 2017*). This motif was detected in 52 of the 187 genera examined, predominantly for genera in the *Proteobacteria*, *Bacteroidetes* and *Cyanobacteria* phyla. The *boxA* consensus is known to vary between species, diverging from the *E. coli* sequence with increasing evolutionary distance (*Arnvig et al., 2008*). Hence, it is possible that *boxA* sequences were missed upstream of CRISPR arrays in bacterial genera that are less closely related to *E. coli*. Given that the

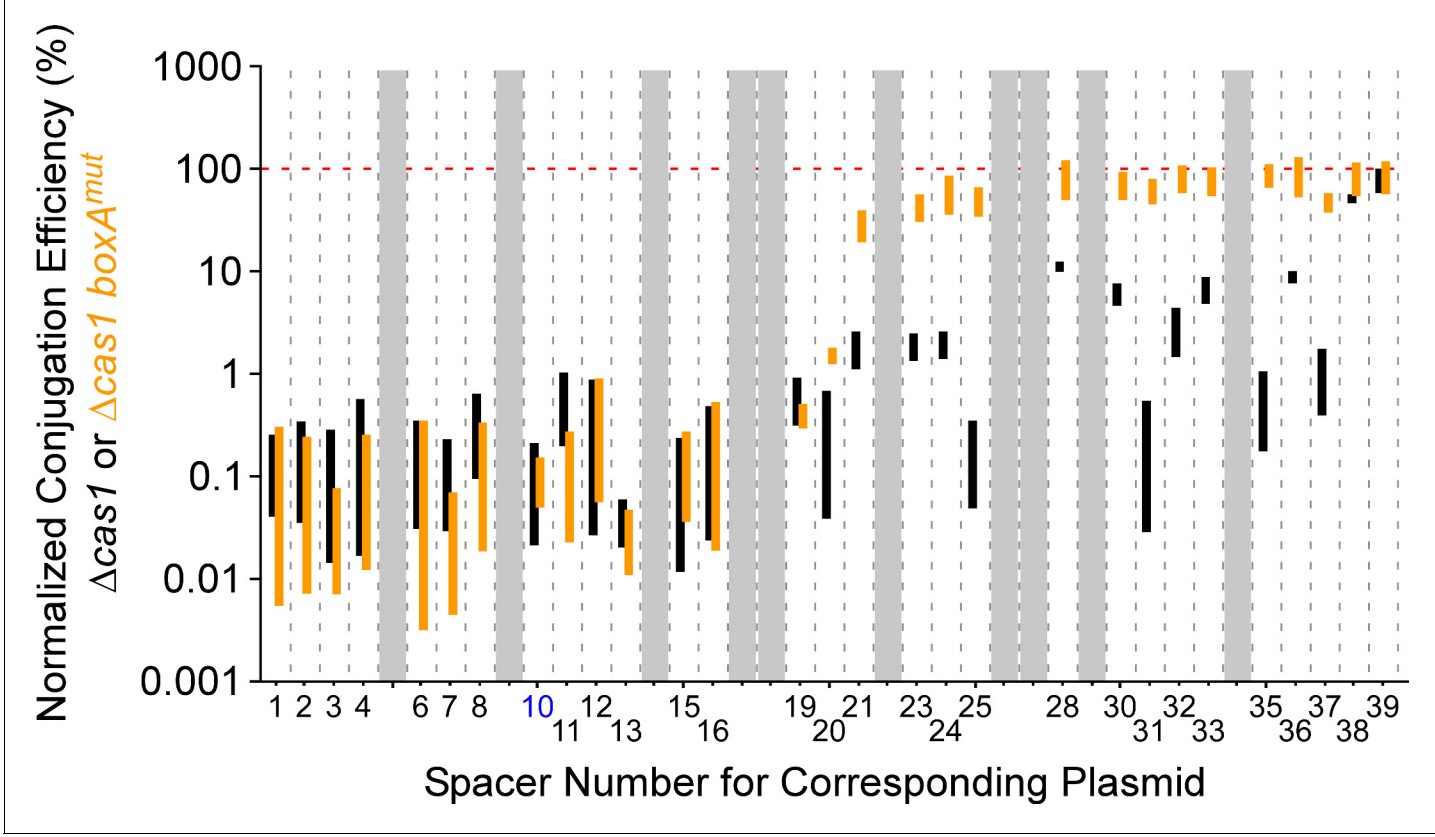

**Figure 6.** Conjugation efficiencies of protospacer-containing plasmids into *Vibrio cholerae*. Normalized conjugation efficiency (%) of protospacer-containing plasmids into Δ*cas1* (KS2094) or *boxA*^mut Δ*cas1* (KS2206) *V. cholerae* A50. Bars indicate the range of conjugation efficiencies across two replicate experiments, with the spacer corresponding to each plasmid indicated on the *x*-axis. Black and orange bars indicate efficiency values for the Δ*cas1* strain and *boxA*^mut Δ*cas1* strain, respectively. Note that spacers 10 and 26 are identical, with data only shown for spacer 10 (blue text). The red horizontal, dashed line indicates 100% conjugation efficiency (i.e. the same efficiency as the control plasmid that lacks a protospacer). Greyed-out regions indicate protospacer-containing plasmids with <50 sequence reads in the control sample for at least one of the two replicates.
The online version of this article includes the following figure supplement(s) for figure 6:

**Figure supplement 1.** Sequence of the CRISPR array and upstream sequence from *Vibrio cholerae* A50.

*Proteobacteria* likely share a more recent common ancestor with phyla other than the *Bacteroidetes* and *Cyanobacteria* (**Bern and Goldberg, 2005**), we propose that *boxA* sequences upstream of CRISPR arrays either (i) evolved in a very early bacterial ancestor but were not detected in our analysis due to *boxA* sequence divergence, or (ii) evolved independently in multiple bacterial lineages. Regardless of its evolutionary history, it is clear that this phenomenon is distributed broadly across the bacterial kingdom.

Conservation of *boxA* sequences upstream of CRISPR arrays in cyanobacterial species is unexpected because these species lack *rho* (**D'Heygère et al., 2013**). The putative *boxA* sequences upstream of cyanobacterial CRISPR arrays differ from the *boxA* consensus sequence derived from *E. coli* (**Figure 7—figure supplement 2**). As discussed above, this could reflect divergence of NusB/E sequence specificity, or may be an indication that these putative *boxA* sequences are false positives. We reasoned that rRNA loci in cyanobacteria would have *boxA* sequences upstream, and that these could be used to determine if the *boxA* consensus sequence in cyanobacteria differs from that in *E. coli*. We searched for enriched sequence motifs in rRNA upstream sequences from the nine species for which we identified putative *boxA* sequences upstream of CRISPR arrays. The most strongly enriched sequence motif from the rRNA upstream regions is a close match to the putative *boxA* sequences upstream of CRISPR arrays from the same species (**Figure 7—figure supplement 2**), and there were no enriched sequence motifs that were more similar to the *E. coli boxA* consensus. We conclude that the *boxA* sequences upstream of cyanobacterial CRISPR arrays are genuine.

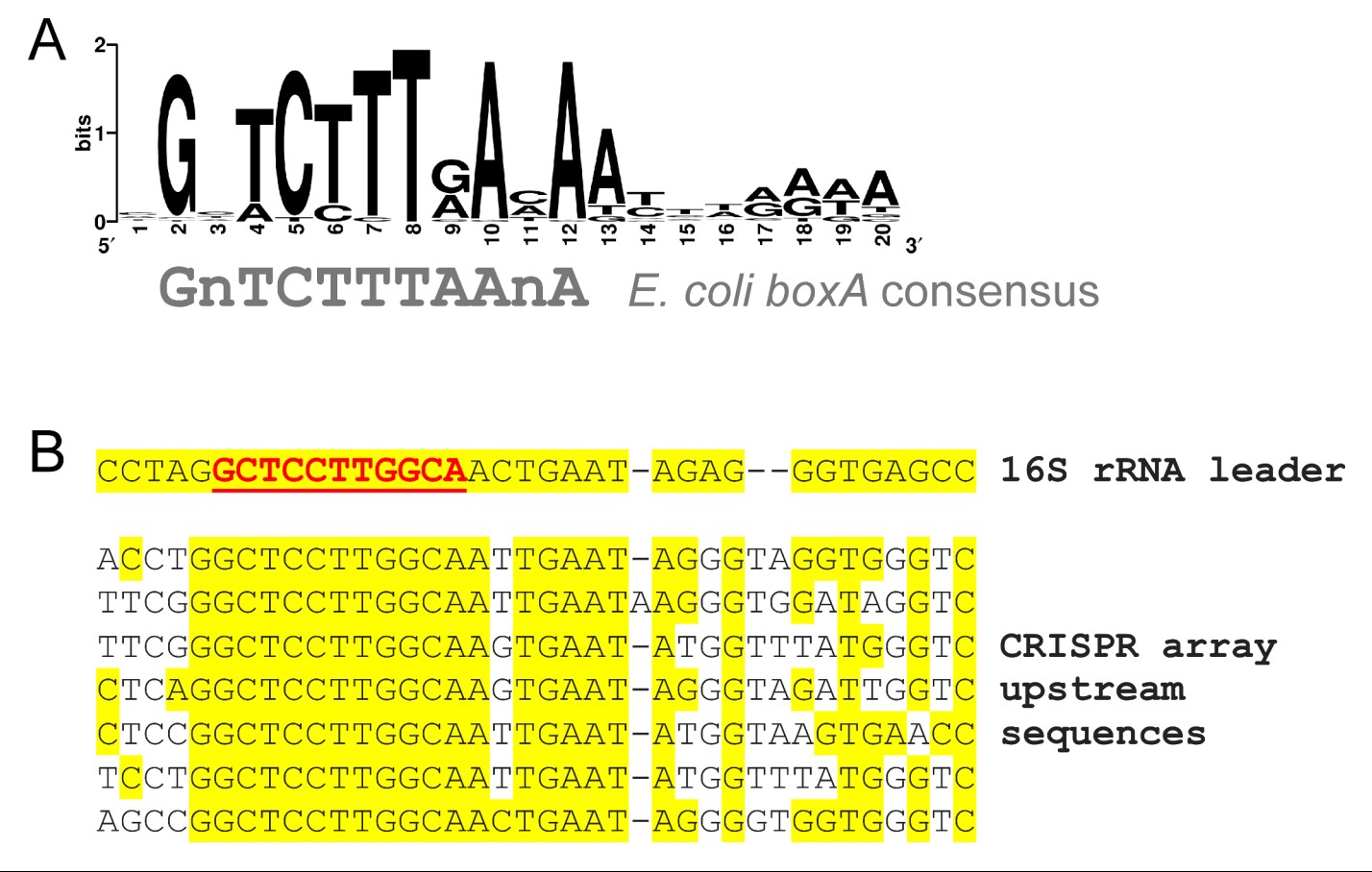

**Figure 7.** Widespread conservation of *boxA* sequences upstream of CRISPR arrays in diverse bacterial genera. (**A**) Sequence motif found by MEME (*Bailey and Elkan, 1994*) to be significantly enriched upstream of 52 CRISPR arrays spread across the bacterial kingdom (from 187 tested; MEME *E*-value = $7.1e^{-69}$). (**B**) A sequence alignment of sequences upstream of CRISPR arrays and ribosomal RNA genes in *Aquifex aeolicus*. The known *boxA* sequence upstream of ribosomal RNA genes is underlined and in red text. Yellow shading indicates identical bases.

The online version of this article includes the following figure supplement(s) for figure 7:

**Figure supplement 1.** Conservation of *nusB* across the bacterial kingdom.

**Figure supplement 2.** Analysis of putative *boxA* sequences upstream of cyanobacterial CRISPR arrays and ribosomal RNA loci.

## Discussion

### Rho poses a barrier to CRISPR array transcription

Our data highlight the importance of transcription antitermination in the process of CRISPR biogenesis. Rho termination is a widely conserved, and often essential process in bacteria. Given that CRISPR arrays and the upstream leader sequences are non-coding, they represent likely substrates for Rho. Thus, Rho may be an unavoidable barrier to expression of longer CRISPR arrays, necessitating an antitermination activity. Although Rho has some specificity for sites of loading and termination, newly acquired spacers in a CRISPR array have sequences that cannot be 'anticipated'; a 'bad' spacer, that is one that promotes Rho loading or termination, could inactivate a large portion of the array by causing premature Rho termination. This phenomenon is exacerbated by the fact that new spacers are added to the upstream end of the array with respect to the direction of transcription. Since spacers only impact cell fitness when the cognate invader is encountered, a bad spacer could be acquired in a CRISPR array and become fixed in the population, inactivating later spacers in the array before cells require immunity using those spacers. We speculate that Rho termination acts as a selective pressure to limit adaptation in species that lack an antitermination mechanism, perhaps explaining why most CRISPR arrays have only a few spacers (*Weissman et al., 2018*). The spacers

most affected by Rho termination will be at the leader-distal ends of CRISPR arrays, meaning that they are the 'older' spacers. This may reduce the selective pressure to prevent Rho termination if the immunity provided by these older spacers is not beneficial due to a reduced likelihood of encountering the cognate invader.

## BoxA-mediated antitermination is a phylogenetically widespread mechanism to prevent premature Rho-dependent transcription termination of CRISPR arrays

We have identified a widely conserved mechanism to counteract premature Rho termination of CRISPR arrays: BoxA-mediated antitermination. The phylogenetic distribution of CRISPR arrays with an associated *boxA* suggests that this mechanism of CRISPR array antitermination has evolved independently on multiple occasions (*Supplementary file 4*). Moreover, BoxA-mediated antitermination of CRISPR arrays is likely to be even more widely distributed than our analysis suggests, because *boxA* sequences that diverge from the *E. coli* consensus may have been missed. Indeed, previous studies of BoxA-mediated antitermination have shown that the *boxA* consensus varies across the bacterial kingdom (*Arnvig et al., 2008*; *Das et al., 2008*). This is highlighted by the fact that, in a separate analysis, we identified putative *boxA* sequences upstream of five CRISPR arrays in *Aquifex aeolicus* (*Figure 7B*). Prediction of these *boxA* sequences, which differ considerably from those in *E. coli* and *S.* Typhimurium, was only possible because of prior experimental work identifying the rRNA *boxA* in *A. aeolicus* (*Das et al., 2008*).

Our data suggest that even in the presence of an antitermination mechanism, Rho termination can still impact CRISPR array transcription. Specifically, we showed that inhibition of Rho led to increased use of later spacers in the two *S.* Typhimurium CRISPR arrays even in the presence of an intact *boxA* (*Figure 4—figure supplement 1*). Similarly, the last two spacers of the *V. cholerae* CRISPR array failed to elicit interference even in wild-type cells (*Figure 6*), suggesting that Rho termination occurs before the end of the array. Thus, it seems likely that even with an antitermination mechanism in place, Rho termination can limit the effective length of CRISPR arrays.

Curiously, there appear to be *boxA* sequences upstream of CRISPR arrays in several species of the *Cyanobacteria* phylum (*Supplementary file 4*; *Figure 7—figure supplement 2*), despite almost all species in this phylum lacking *rho* (*D'Heygère et al., 2013*). We propose two possible explanations. First, there may be an alternative transcription termination mechanism in cyanobacteria that can be antagonized by the Nus factor complex. Second, Nus factor association with RNAP transcribing a CRISPR array likely generates an RNA loop, since the BoxA is tethered to RNAP by the Nus factors. Loop formation could impact crRNA processing by modulating base-pairing interactions in the nascent RNA. BoxA-mediated RNA loop formation has been proposed to facilitate folding and processing of rRNA (*Bubunenko et al., 2013*; *Singh et al., 2016*). Such a function need not be mutually exclusive with transcription antitermination.

## Are there alternative antitermination mechanisms for CRISPR arrays?

Our search for enriched sequence motifs upstream of CRISPR arrays used arrays from 187 bacterial species. Most of these species have type I CRISPR-Cas systems, often in combination with additional types. Fifty of the 52 species with a putative *boxA* sequence upstream of a CRISPR array have at least one type I system, with the other two species each having only a type III system. By contrast, none of the 15 species with only a type II-C CRISPR-Cas system have *boxA* sequences upstream of their arrays. We reasoned that this might be because at least some species with type II-C CRISPR-Cas systems have their CRISPR arrays oriented opposite to *cas2*, rather than co-directionally (*Dugar et al., 2013*; *Zhang et al., 2013*); when extracting sequences adjacent to CRISPR arrays for sequence analysis (*Figure 7A*), we assumed that *cas2* and CRISPR arrays are oriented co-directionally. Indeed, it is possible that *boxA* sequences were missed for other types of CRISPR array for the same reason. To test the possibility that there are *boxA* sequences on the *cas2*-distal side of the 15 type II-C CRISPR arrays included in our analysis, we replaced the sequences between *cas2* and CRISPR arrays with the 300 nt from the opposite end of the CRISPR arrays for all 15 species, and repeated the search for enriched sequence motifs; sequences used for the other 172 species remained the same. We did not detect a putative *boxA* adjacent to any of the type II-C arrays (*Supplementary file 5*), suggesting that these species use a different mechanism to circumvent Rho

termination within their CRISPR arrays. Indeed, repeat sequences in many type II-C CRISPR-Cas systems contain promoters, such that each spacer in the array is transcribed from a promoter immediately upstream (*Charpentier et al., 2015*; *Dugar et al., 2013*; *Mir et al., 2018*; *Zhang et al., 2013*). We propose that this provides an alternative mechanism to avoid Rho termination, obviating the need for a *boxA* sequence.

Given that many bacterial CRISPR arrays lack an upstream *boxA*, even in species that are closely related to *Salmonella*, we anticipate that there are additional mechanisms of CRISPR array antitermination. These are likely to be processive antitermination mechanisms, since targeted antitermination would be incompatible with the dynamic nature of CRISPR arrays. Nonetheless, we note that targeted antitermination mechanisms may be important for preventing Rho termination in the leader sequences upstream of CRISPR arrays, which are often long and non-coding. Indeed, a targeted antitermination mechanism has been described for CRISPR array leader sequences in *Pseudomonas aeruginosa* (*Lin et al., 2019*). Alternatives to BoxA-mediated antitermination of CRISPR arrays could include antitermination by NusG homologues such as RfaH (*Goodson et al., 2017*; *Goodson and Winkler, 2018*; *Kang et al., 2018*), or inhibition of Rho activity by *cis*-acting RNA elements, similar to the Put element of phage HK022 (*King and Weisberg, 2003*). An alternative possibility is that CRISPR array processing by Cas6 may be so efficient in some bacterial species that the CRISPR array transcript is processed downstream of Rho, preventing Rho from catching RNAP on the RNA. Lastly, there may be mechanisms to circumvent Rho termination that are independent of processive antitermination, such as the repeat-internal promoters found in type II-C CRISPR-Cas systems (*Charpentier et al., 2015*; *Dugar et al., 2013*; *Mir et al., 2018*; *Zhang et al., 2013*).

## Conclusions

In conclusion, we have identified Rho termination as an important player in CRISPR biogenesis, and we have identified a widespread mechanism to counteract premature Rho termination of CRISPR arrays. We anticipate that studies of CRISPR biogenesis in other bacterial species will reveal novel antitermination mechanisms, and that bacteriophage have evolved mechanisms to counteract antitermination as an anti-CRISPR strategy. Lastly, the potential for Rho termination should be considered for biotechnological applications that require arrays with multiple spacers. Indeed, a recent study showed that including a *boxA* upstream of a 10-spacer CRISPR array substantially improves the efficiency of multiplexed CRISPRi in *Legionella pneumophila* (*Ellis et al., 2020*).

# Materials and methods

**Key resources table**

| Reagent type (species) or resource | Designation | Source or reference | Identifiers | Additional information |
|---|---|---|---|---|
| Strain (*Salmonella* Typhimurium) | 14028s | DOI:10.1128/JB.01233–09 | | |
| Strain (*Salmonella* Typhimurium) | AMD678 | This study | 14028s PKAB-TG::Δ*cas3*, FLAG3-*cas5*, P$_{KAB-TG}$::CRISPR-II | *Supplementary file 6* |
| Strain (*Salmonella* Typhimurium) | AMD679 | This study | AMD678 ΔCRISPR-I::*thyA* | *Supplementary file 6* |
| Strain (*Salmonella* Typhimurium) | AMD684 | This study | AMD678 CRISPR-I boxA(C4A) | *Supplementary file 6* |
| Strain (*Salmonella* Typhimurium) | AMD685 | This study | AMD678 CRISPR-II boxA(C4A) | *Supplementary file 6* |

*Continued on next page*

*Continued*

| Reagent type (species) or resource | Designation | Source or reference | Identifiers | Additional information |
|---|---|---|---|---|
| Strain (*Salmonella* Typhimurium) | AMD698 | This study | AMD678 nusE(N3H) | *Supplementary file 6* |
| Strain (*Salmonella* Typhimurium) | AMD710 | This study | 14028s P$_{KAB-TG}$::Δcas3, FLAG3-*cas5*, *suhB*-TAP, P$_{J23119}$-*thyA*::CRISPR-II | *Supplementary file 6* |
| Strain (*Salmonella* Typhimurium) | AMD711 | This study | AMD710 CRISPR-II *boxA*(C4A) | *Supplementary file 6* |
| Strain (*Vibrio cholerae*) | KS1234 | DOI: 10.1128/ JB.00747–15 | | |
| Strain (*Vibrio cholerae*) | KS2094 | This study | KS1234 Δcas1 | *Supplementary file 6* |
| Strain (*Vibrio cholerae*) | KS2206 | This study | KS1234 Δcas1 *boxAmut* (C4A, T5G) | *Supplementary file 6* |
| Strain (*Vibrio cholerae*) | BJO185 | DOI: 10.1128/ JB.00747–15 | | |
| Strain (*Escherichia coli*) | S17 | DOI: 10.1016/ 0378-1119(90) 90147-J | | |
| Antibody | Anti-*E. coli* RNA Polymerase β Antibody (Mouse monoclonal) | BioLegend | Catalog # 663903 | ChIP (1 µL) |
| Antibody | Monoclonal anti-FLAG M2 antibody (Mouse monoclonal) | SIGMA | Catalog # F1804 | ChIP-seq (2 µL) |
| Recombinant DNA reagent | pBAD24- amp (plasmid) | DOI: 10.1128/ jb.177.14.4121– 4130.1995 | | |
| Recombinant DNA reagent | pAMD239 (plasmid) | This study | pBAD24-*nusE* | |
| Recombinant DNA reagent | pGEM-T (plasmid) | Promega, Catalog # A1360 | | |
| Recombinant DNA reagent | pVS030 (plasmid) | This study | pGEM-T-TAP-*thyA*-TAP | |
| Recombinant DNA reagent | pJTW064 (plasmid) | DOI: 10.1128/ JB.01007–13 | | |
| Recombinant DNA reagent | pGB231 (plasmid) | This study | pJTW064 with CRISPR-II upstream region and up to spacer 2 | *Supplementary file 6* |
| Recombinant DNA reagent | pGB237 (plasmid) | This study | pJTW064 with CRISPR-II upstream region and up to spacer 2, *boxA*(C4A) | *Supplementary file 6* |
| Recombinant DNA reagent | pGB250 (plasmid) | This study | pJTW064 with CRISPR-II upstream region and up to spacer 11 | *Supplementary file 6* |

*Continued on next page*

*Continued*

| Reagent type (species) or resource | Designation | Source or reference | Identifiers | Additional information |
|---|---|---|---|---|
| Recombinant DNA reagent | pGB256 (plasmid) | This study | pJTW064 with CRISPR-II upstream region and up to spacer 11, *boxA*(C4A) | *Supplementary file 6* |
| Recombinant DNA reagent | p958 (plasmid) | DOI: 10.1128/ JB.00747–15 | | |
| Recombinant DNA reagent | pAMD244 (plasmid) | This study | pKS958 with *V. cholerae* A50 protospacer 1 | *Supplementary file 6* |
| Recombinant DNA reagent | pAMD245 (plasmid) | This study | pKS958 with *V. cholerae* A50 protospacer 2 | *Supplementary file 6* |
| Recombinant DNA reagent | pAMD246 (plasmid) | This study | pKS958 with *V. cholerae* A50 protospacer 3 | *Supplementary file 6* |
| Recombinant DNA reagent | pAMD247 (plasmid) | This study | pKS958 with *V. cholerae* A50 protospacer 4 | *Supplementary file 6* |
| Recombinant DNA reagent | pAMD248 (plasmid) | This study | pKS958 with *V. cholerae* A50 protospacer 5 | *Supplementary file 6* |
| Recombinant DNA reagent | pAMD249 (plasmid) | This study | pKS958 with *V. cholerae* A50 protospacer 6 | *Supplementary file 6* |
| Recombinant DNA reagent | pAMD250 (plasmid) | This study | pKS958 with *V. cholerae* A50 protospacer 7 | *Supplementary file 6* |
| Recombinant DNA reagent | pAMD251 (plasmid) | This study | pKS958 with *V. cholerae* A50 protospacer 8 | *Supplementary file 6* |
| Recombinant DNA reagent | pAMD252 (plasmid) | This study | pKS958 with *V. cholerae* A50 protospacer 9 | *Supplementary file 6* |
| Recombinant DNA reagent | pAMD253 (plasmid) | This study | pKS958 with *V. cholerae* A50 protospacer 11 | *Supplementary file 6* |
| Recombinant DNA reagent | pAMD254 (plasmid) | This study | pKS958 with *V. cholerae* A50 protospacer 12 | *Supplementary file 6* |
| Recombinant DNA reagent | pAMD255 (plasmid) | This study | pKS958 with *V. cholerae* A50 protospacer 13 | *Supplementary file 6* |
| Recombinant DNA reagent | pAMD256 (plasmid) | This study | pKS958 with *V. cholerae* A50 protospacer 14 | *Supplementary file 6* |
| Recombinant DNA reagent | pAMD257 (plasmid) | This study | pKS958 with *V. cholerae* A50 protospacer 15 | *Supplementary file 6* |
| Recombinant DNA reagent | pAMD258 (plasmid) | This study | pKS958 with *V. cholerae* A50 protospacer 16 | *Supplementary file 6* |
| Recombinant DNA reagent | pAMD259 (plasmid) | This study | pKS958 with *V. cholerae* A50 protospacer 17 | *Supplementary file 6* |

*Continued on next page*

*Continued*

| Reagent type (species) or resource | Designation | Source or reference | Identifiers | Additional information |
|---|---|---|---|---|
| Recombinant DNA reagent | pAMD260 (plasmid) | This study | pKS958 with *V. cholerae* A50 protospacer 19 | *Supplementary file 6* |
| Recombinant DNA reagent | pAMD262 (plasmid) | This study | pKS958 with *V. cholerae* A50 protospacer 20 | *Supplementary file 6* |
| Recombinant DNA reagent | pAMD263 (plasmid) | This study | pKS958 with *V. cholerae* A50 protospacer 21 | *Supplementary file 6* |
| Recombinant DNA reagent | pAMD264 (plasmid) | This study | pKS958 with *V. cholerae* A50 protospacer 22 | *Supplementary file 6* |
| Recombinant DNA reagent | pAMD265 (plasmid) | This study | pKS958 with *V. cholerae* A50 protospacer 23 | *Supplementary file 6* |
| Recombinant DNA reagent | pAMD266 (plasmid) | This study | pKS958 with *V. cholerae* A50 protospacer 24 | *Supplementary file 6* |
| Recombinant DNA reagent | pAMD267 (plasmid) | This study | pKS958 with *V. cholerae* A50 protospacer 25 | *Supplementary file 6* |
| Recombinant DNA reagent | pAMD268 (plasmid) | This study | pKS958 with *V. cholerae* A50 protospacer 10/26 | *Supplementary file 6* |
| Recombinant DNA reagent | pAMD269 (plasmid) | This study | pKS958 with *V. cholerae* A50 protospacer 27 | *Supplementary file 6* |
| Recombinant DNA reagent | pAMD270 (plasmid) | This study | pKS958 with *V. cholerae* A50 protospacer 28 | *Supplementary file 6* |
| Recombinant DNA reagent | pAMD271 (plasmid) | This study | pKS958 with *V. cholerae* A50 protospacer 29 | *Supplementary file 6* |
| Recombinant DNA reagent | pAMD272 (plasmid) | This study | pKS958 with *V. cholerae* A50 protospacer 30 | *Supplementary file 6* |
| Recombinant DNA reagent | pAMD273 (plasmid) | This study | pKS958 with *V. cholerae* A50 protospacer 31 | *Supplementary file 6* |
| Recombinant DNA reagent | pAMD274 (plasmid) | This study | pKS958 with *V. cholerae* A50 protospacer 32 | *Supplementary file 6* |
| Recombinant DNA reagent | pAMD275 (plasmid) | This study | pKS958 with *V. cholerae* A50 protospacer 33 | *Supplementary file 6* |
| Recombinant DNA reagent | pAMD276 (plasmid) | This study | pKS958 with *V. cholerae* A50 protospacer 34 | *Supplementary file 6* |
| Recombinant DNA reagent | pAMD277 (plasmid) | This study | pKS958 with *V. cholerae* A50 protospacer 35 | *Supplementary file 6* |
| Recombinant DNA reagent | pAMD278 (plasmid) | This study | pKS958 with *V. cholerae* A50 protospacer 36 | *Supplementary file 6* |

*Continued on next page*

*Continued*

| Reagent type (species) or resource | Designation | Source or reference | Identifiers | Additional information |
|---|---|---|---|---|
| Recombinant DNA reagent | pAMD279 (plasmid) | This study | pKS958 with *V. cholerae* A50 protospacer 37 | *Supplementary file 6* |
| Recombinant DNA reagent | pAMD280 (plasmid) | This study | pKS958 with *V. cholerae* A50 protospacer 38 | *Supplementary file 6* |
| Recombinant DNA reagent | pAMD281 (plasmid) | This study | pKS958 with *V. cholerae* A50 protospacer 39 | *Supplementary file 6* |
| Chemical compound, drug | Bicyclomycin | Santa Cruz Biotechnology, Inc | Catalog # sc-391755 | |
| Chemical compound | Protein A Sepharose CL-4B Medium | Cytiva (Formerly GE Healthcare Life Sciences) | Catalog # 17078001 | |
| Chemical compound | IgG Sepharose 6 Fast Flow Medium | Cytiva (Formerly GE Healthcare Life Sciences) | Catalog # 17096901 | |

## Strains and plasmids

All strains, plasmids, and oligonucleotides used in this study are listed in *Supplementary files 6* and *7*, respectively. All *Salmonella* strains are derivatives of *Salmonella enterica* subspecies *enterica* serovar Typhimurium 14028s (*Jarvik et al., 2010*). Strains AMD678, AMD679, AMD684, AMD685, AMD698, AMD710, and AMD711 were generated using the FRUIT recombineering method (*Stringer et al., 2012*). To construct AMD678, oligonucleotides JW8576 and JW8577 were used to N-terminally FLAG$_3$-tag *cas5*. Then, oligonucleotides JW8610 + JW8611 and JW8797 + JW8798 were used for insertion of a constitutive P$_{KAB-TG}$ promoter (*Burr et al., 2000*) in place of *cas3*, and upstream of the CRISPR-II array, respectively. AMD679, AMD684, AMD685, and AMD698 are derivatives of AMD678 and were generated using oligonucleotides (i) JW8913 and JW8914 to amplify *thyA* and replace the CRISPR-I array, (ii) JW8904-JW8907 for introduction of a CRISPR-I *boxA* mutation (C4A), (iii) JW8568-JW8571 for introduction of a CRISPR-II *boxA* mutation (C4A), and (iv) JW9441-JW9444 for introduction of a *nusE* mutation (N3H). Mutation of *nusE* was associated with deletion of a 146.6 kb region, from genome position 1654295 to 1800903 (determined from ChIP-seq reads spanning the deletion). It is unclear whether this deletion is required for viability. AMD710 and AMD711 were derived from AMD678 and AMD685, respectively. They were constructed using oligonucleotides JW9355 + JW9356 to C-terminally TAP-tag *suhB*, with pVS030 (see below) as a PCR template. The constitutive P$_{KAB-TG}$ promoter (*Burr et al., 2000*) upstream of the CRISPR-II array was replaced by P$_{J23119}$-*thyA*, using oligonucleotides JW9627 + JW9628 and a strain containing *thyA* driven by the P$_{J23119}$ promoter as a template for colony PCR.

All *V. cholerae* strains are derivatives of A50 (*Mutreja et al., 2011*). In-frame unmarked deletions and point mutations were constructed using splicing by overlap extension (SOE) PCR (*Horton et al., 1989*) and introduced through conjugation with *E. coli* SM10λpir using pCVD442-*lac* (*Donnenberg and Kaper, 1991*). Constructs were generated using oligonucleotides (i) KS1297-KS1300 for introduction of a 672 bp in-frame deletion in *cas1* (leaving a 204 bp non-functional allele), and (ii) KS1368-KS1371 for introduction of the *boxA* mutation (C4A, T5G).

The plasmid pAMD239 was constructed by using oligonucleotides JW9674 and JW9675 to amplify *nusE* from wild-type 14028s. The resulting DNA fragment was cloned into pBAD24 (*Guzman et al., 1995*) digested with HindIII and NheI. For the construction of pVS030, duplicate sets of TAP tags were colony PCR-amplified from a TAP-tagged strain of *E. coli* (*Butland et al., 2005*) using oligonucleotides JW6401 + JW6445 and JW6448 + JW6406. Oligonucleotides JW6446

+ JW6447 were used to colony PCR-amplify *thyA*, as described previously (*Stringer et al., 2012*). All three DNA fragments were cloned into the pGEM-T plasmid (Promega) digested with *Sal*I and *Nco*I.

Plasmids pGB231, pGB237, pGB250, pGB256 were constructed by PCR-amplifying and cloning a truncated *S*. Typhimurium 14028s CRISPR-II array (to the 2<sup>nd</sup> or 11<sup>th</sup> spacer) and 292 bp of upstream sequence into the *Nsi*I and *Nhe*I sites of the pJTW064 plasmid (*Stringer et al., 2014*) with either a wild-type or mutant *boxA* sequence. A constitutive promoter was introduced upstream of the CRISPR-II sequence, creating a transcriptional fusion to *lacZ*. The following primers were used to PCR-amplify each CRISPR-II truncation: JW9381 + JW9383 (pGB231 and pGB237), JW9381 + JW9605 (pGB250 and pGB256). Fusions carrying *boxA* mutations were made by amplifying CRII from AMD678 template, which contains a *boxA*(C4A) mutation.

Protospacer plasmids pAMD244-pAMD260 and pAMD262-pAMD281 were constructed by cloning each spacer from the *V. cholerae* A50 CRISPR array into *Hind*III-digested pKS958 (*Box et al., 2016*) using In-Fusion (Clonetech). The protospacer plasmid matching array spacer 18 contained a point mutation and was discarded.

## ChIP-qPCR

Strains 14028s, AMD678, AMD710, and AMD711 were subcultured 1:100 in LB and grown to an $OD_{600}$ of 0.5–0.8. ChIP and input samples were prepared and analyzed as described previously (*Stringer et al., 2014*). For ChIP of β, 1 μl anti-β (RNA polymerase subunit) antibody was used. For ChIP of SuhB-TAP, IgG Sepharose was used in place of Protein A Sepharose. Enrichment of ChIP samples was determined using quantitative real-time PCR with an ABI 7500 Fast instrument, as described previously (*Aparicio et al., 2005*). Enrichment was calculated relative to a control region, within the *sseJ* gene (PCR-amplified using oligonucleotides JW4477 + JW4478), which is expected to be free of SuhB and RNA polymerase. Oligonucleotides used for qPCR amplification of the region within the CRISPR-I array were JW9305 + JW9306. Oligonucleotides used for qPCR amplification of the region surrounding the CRISPR-II *boxA* were JW9329 + JW9330. Oligonucleotides used for qPCR amplification of the region within *rpsA* were JW9660 + JW9661. Occupancy values represent background-subtracted enrichment relative to the control region.

## ChIP-seq

All ChIP-seq experiments were performed in duplicate. Cultures were inoculated 1:100 in LB with fresh overnight cultures of AMD678, AMD679, AMD684, and AMD685. Cultures of AMD678, AMD684, and AMD685 were split into two cultures at an $OD_{600}$ of 0.1, and bicyclomycin was added to one of the two cultures to a concentration of 20 μg/mL. At an $OD_{600}$ of 0.5–0.8, cells were processed for ChIP-seq of FLAG₃-Cas5, following a protocol described previously (*Stringer et al., 2014*). For ChIP-seq using derivatives of the *nusE* mutant strain (AMD698), AMD698 containing either empty pBAD24 or pAMD239, was subcultured 1:100 in LB supplemented with 100 μg/mL ampicillin and 0.2% arabinose. AMD698 + pBAD24 cultures were split into two cultures at an $OD_{600}$ of 0.1. Bicyclomycin was added to one of the two cultures to a concentration of 20 μg/mL. At an $OD_{600}$ of 0.5–0.8, cells were processed for ChIP-seq of FLAG₃-Cas5, following a protocol described previously (*Stringer et al., 2014*).

## ChIP-seq data analysis

Peak calling from ChIP-seq data was performed as previously described (*Fitzgerald et al., 2014*). To assign specific crRNA spacer sequences to Cascade-binding sites, we extracted 101 bp regions centered on each ChIP-seq peak for ChIP-seq data generated from AMD678. Overlapping regions were merged and the central position was used as a reference point for downstream analysis. We refer to this position as the 'peak center'. We then searched each ChIP-seq peak region for a perfect match to positions 1–5 of each spacer from the CRISPR-I and CRISPR-II arrays, in addition to an immediately adjacent AAG or ATG sequence (the expected PAM sequence). Additionally, we searched each ChIP-seq peak region for a perfect match to positions 1–5 and positions 7–8 of each spacer from the CRISPR-I and CRISPR-II arrays without an associated PAM. Spacers were only assigned to a ChIP-seq peak if they had a unique match to a spacer sequence. This yielded 152 uniquely assigned peak-spacer combinations from the 236 ChIP-seq peaks. Enriched sequence motifs within the 236

peak regions (*Figure 3B*) were identified using MEME (v5.0.1, default parameters) using 101 bp sequences surrounding peak centers (*Bailey and Elkan, 1994*).

To determine relative sequence read coverage at each ChIP-seq peak center, we used Rockhopper (*McClure et al., 2013*) to determine relative sequence coverage at every genomic position on both strands for each ChIP-seq dataset. We then summed the relative sequence read coverage values on both strands for each peak center position to give peak center coverage values (*Supplementary file 2*). To refine the assignment of spacers to ChIP-seq peaks, we compared peak center coverage values for each peak in ChIP-seq datasets from AMD678 (CRISPR-I$^+$) and AMD679 (ΔCRISPR-I) strains. We calculated ratio of peak center coverage values in the first replicates of AMD679:AMD678 data, and repeated this for the second replicates, generating two ratio values. Based on ratio values assigned to peak centers that were already uniquely assigned to a spacer, we conservatively assumed that peak centers with both ratios < 0.2 should be assigned a CRISPR-I spacer, whereas peak centers with both ratios > 1.0 should be assigned a CRISPR-II spacer. Thus, we were able to uniquely assign spacers to an additional 32 peak centers that had previously been assigned multiple spacers. For example, if a peak center had previously been assigned two spacers from CRISPR-I and one spacer from CRISPR-II, we could uniquely assign the spacer from CRISPR-II if both ratios were >1.0.

For data plotted in *Figures 4* and *5*, S2, S3, and S4, values for peak center coverage were normalized in one replicate (two biological replicates were performed for all ChIP-seq experiments) by summing the values at all peak centers to be analyzed (i.e. peak centers that could be uniquely assigned to a spacer) and multiplying values in the second replicate by a constant such that the summed values for each replicate were the same.

## β-Galactosidase assays

*S.* Typhimurium 1402814028s containing pGB231, pGB237, pGB250, pGB256, or pJTW060 was grown at 37°C in LB medium to an OD$_{600}$ of 0.4–0.6. 810 µL of culture was pelleted and resuspended in 810 µL of Z buffer (0.06 M Na$_2$HPO$_4$, 0.04 M NaH$_2$PO$_4$, 0.01 M KCl, 0.001 M MgSO$_4$) + 50 mM β-mercaptoethanol + 0.001% SDS + 20 µL chloroform. Cells were lysed by brief vortexing. β-Galactosidase reactions were started by adding 160 µL 2-Nitrophenyl β-D-galactopyranoside (4 mg/mL) and stopped by adding 400 µL of 1 M Na$_2$CO$_3$. The duration of the reaction and OD$_{420}$ readings were recorded. β-galactosidase activity was calculated as 1000 * (A$_{420}$/(A$_{600}$)(time [min] * volume [mL])).

## High-throughput assessment of plasmid conjugation into *Vibrio cholerae*

Protospacer plasmids and an empty pKS958 control were transformed into *E. coli* S17 (*Parke, 1990*). All *E. coli* S17 strains containing protospacer plasmids were grown at 37°C in LB supplemented with 30 µg/mL chloramphenicol to an OD$_{600}$ of 0.3–1.0, and then pooled proportionally, with the exception of the cultures harboring protospacer plasmids 38, 39, and empty pKS958, where only 1/10th of the amount of cells was added to the pool. The plasmid pool was transferred to *V. cholerae* strains BJO185, KS2094, and KS2206 by conjugation, as previously described (*Box et al., 2016*). Following plasmid transfer, 100 µL of the cell mixture from each conjugation sample was plated at 37°C on LB agar supplemented with 75 µg/mL kanamycin and 2.5 µg/mL chloramphenicol. The next day, colonies were scraped from the plates. In the second replicate, scraped cells were resuspended in 10 mL M9 minimal medium and pelleted by centrifugation. In the first replicate, scraped cells were resuspended in LB to an OD$_{600}$ of 0.1, grown without antibiotic selection at 37°C with shaking for 4 hr and pelleted. ~1 µL of cells from each pellet added to 20 µL dH2O and incubated at 95 °C for 10 min. From each cell resuspension, 1.0 µL was used as a template in a PCR reaction with universal forward oligonucleotide JW10344 and reverse index oligonucleotides JW10345 for BJO185 template, JW10346 for KS2094 template, and JW10347 for KS2206 templates. Sequencing was performed using an Illumina Next-Seq instrument (Wadsworth Center Applied Genomic Technologies Core). Sequence reads were assigned to each protospacer-containing plasmid by searching for an exact match to a 10 nt sequence within the protospacer using a custom Python script (*Supplementary file 8*). Any plasmids for which we mapped fewer than 50 sequence reads in the Δcascade sample for either replicate were discarded from the analysis. To calculate normalized conjugation efficiency for each protospacer-containing plasmid into the Δcas1 (*boxA$^+$*) and *boxAmut Δcas1* strains, we first

normalized all sequence read counts to the corresponding value for the Δ*cascade* strain. We then normalized these values to the value for the control plasmid that lacks a protospacer.

### Phylogenetic analysis of CRISPR array boxA conservation

Cas2 protein sequences were extracted from the PF09707 and PF09827 PFAM families (*Finn et al., 2016*; *Sonnhammer et al., 1997*). These sequences were then used to search a local collection of bacterial genome sequences using tBLASTn (*Altschul et al., 1990*). We then selected the 612 sequences with perfect matches to full-length Cas2 sequences. We further refined this set of sequences by arbitrarily selecting only one sequence per genus. We then extracted 300 bp downstream of each *cas2* gene. We trimmed the 3′ end of each sequence at the first instance of a 15 bp repeat, which is likely to be the beginning of a CRISPR array, since *cas2* is often found upstream of a CRISPR array. Any sequences that lacked a 15 bp repeat, or were trimmed to <20 bp, were discarded. We used MEME (v5.1.1, default parameters, except for selecting 'search given strand only') (*Bailey and Elkan, 1994*) to identify enriched sequence motifs in the remaining 187 sequences (*Supplementary file 4*). The type(s) of CRISPR-Cas system found in each species, and sequences flanking CRISPR arrays in species with type II-C CRISPR-Cas systems were determined using the CRISPRCasdb (*Pourcel et al., 2020*) and CRISPRFinder webservers (*Grissa et al., 2007*).

### Phylogenetic analysis of nusB conservation

Conservation of *nusB* across the bacterial kingdom was assessed using the Aquerium tool (*Adebali and Zhulin, 2017*).

### Analysis of sequences flanking CRISPR arrays in *A. aeolicus*

Sequences flanking CRISPR arrays in *A. aeolicus* were extracted using the CRISPRFinder webserver (*Grissa et al., 2007*). Subsequences were aligned with each other and ribosomal RNA leader sequencing using CLUSTAL Omega (*Sievers and Higgins, 2014*).

### Accession numbers

Raw ChIP-seq data are available from EBI ArrayExpress/ENA using accession number E-MTAB-7242. Raw sequencing data for conjugation experiments involving *V. cholerae* are available from EBI ArrayExpress/ENA using accession number E-MTAB-9631.

## Acknowledgements

We thank the Wadsworth Center Applied Genomic Technologies Core Facility for sequencing. We thank the Wadsworth Center Bioinformatics Core Facility for bioinformatic support. We thank the Wadsworth Center Media and Tissue Culture and Glassware Core Facilities. We thank Shailab Shrestha, Todd Gray, Keith Derbyshire, and Randy Morse for helpful discussions. We thank Vimi Singh for creating the TAP-tagging construct used in this study. This study was supported by NIH Grants AI126416 and GM122836 (to JTW) and AI127652 (to KDS). KDS is a Chan Zuckerberg Biohub Investigator and holds an 'Investigators in the Pathogenesis of Infectious Disease' Award from the Burroughs Wellcome Fund.

## Additional information

### Funding

| Funder | Grant reference number | Author |
|---|---|---|
| National Institute of General Medical Sciences | R01GM122836 | Joseph T Wade |
| National Institute of Allergy and Infectious Diseases | R21AI126416 | Joseph T Wade |
| National Institute of Allergy and Infectious Diseases | R01AI127652 | Kimberley D Seed |
| Burroughs Wellcome Fund | Investigators in the | Kimberley D Seed |

Pathogenesis of Infectious
Disease Award

The funders had no role in study design, data collection and interpretation, or the
decision to submit the work for publication.

## Author contributions
Anne M Stringer, Formal analysis, Investigation, Visualization, Writing - review and editing; Gabriele
Baniulyte, Formal analysis, Investigation, Writing - review and editing; Erica Lasek-Nesselquist, For-
mal analysis, Investigation; Kimberley D Seed, Investigation, Writing - review and editing; Joseph T
Wade, Conceptualization, Formal analysis, Supervision, Funding acquisition, Investigation, Writing -
original draft, Project administration, Writing - review and editing

## Author ORCIDs
Gabriele Baniulyte (iD) http://orcid.org/0000-0003-0235-7938
Kimberley D Seed (iD) http://orcid.org/0000-0002-0139-1600
Joseph T Wade (iD) https://orcid.org/0000-0002-9779-3160

## Decision letter and Author response
Decision letter https://doi.org/10.7554/eLife.58182.sa1
Author response https://doi.org/10.7554/eLife.58182.sa2

# Additional files

## Supplementary files
• Supplementary file 1. List of FLAG3-Cas5 ChIP-seq peaks.

• Supplementary file 2. Assignment of ChIP-seq peak centers to spacers, and relative Cas5 occu-
pancy at each peak center.

• Supplementary file 3. Conservation analysis of *nusB* across the bacterial kingdom.

• Supplementary file 4. Sequences used for phylogenetic analysis of CRISPR array upstream regions.

• Supplementary file 5. Sequences used for phylogenetic analysis of CRISPR array upstream regions,
with oppositely oriented sequences for species with just a type II-C CRISPR-Cas system.

• Supplementary file 6. List of strains and plasmids used in this study.

• Supplementary file 7. List of oligonucleotides used in this study.

• Supplementary file 8. Custom python script to count *Vibrio cholerae* protospacers from a fastq file.

• Transparent reporting form

## Data availability
Raw ChIP-seq data are available from EBI ArrayExpress/ENA using accession number E-MTAB-7242.
Raw sequencing data for conjugation experiments involving *V. cholerae* are available from EBI
ArrayExpress/ENA using accession number E-MTAB-9631.

The following datasets were generated:

| Author(s) | Year | Dataset title | Dataset URL | Database and Identifier |
|---|---|---|---|---|
| Stringer AM, Wade JT | 2018 | Nus factors prevent premature transcription termination of bacterial CRISPR arrays | https://www.ebi.ac.uk/arrayexpress/experiments/E-MTAB-7242/ | ArrayExpress, E-MTAB-7242 |
| Stringer AM, Wade JT | 2020 | Transcription Termination and Antitermination of Bacterial CRISPR Arrays | https://www.ebi.ac.uk/arrayexpress/experiments/E-MTAB-9631/ | ArrayExpress, E-MTAB-9631 |

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
