## [Decision Letter]

**Acceptance summary:**

While the mechanisms of CRISPR RNA-guided defense have been the subject of intense investigation, the regulatory mechanisms that govern the transcription of CRISPR loci remains relatively obscure. Here Stringer et al. demonstrate that Rho and Nus play opposing roles in regulating the length of CRISPR transcripts. Longer CRISPR transcripts result in more guides and may provide broad spectrum resistance, but short transcripts may result in higher concentrations of certain guides, providing higher levels of protection from fewer pathogens.

**Decision letter after peer review:**

Thank you for submitting your article "Transcription Termination and Antitermination of Bacterial CRISPR Arrays" for consideration by *eLife*. Your article has been reviewed by three peer reviewers, one of whom is a member of our Board of Reviewing Editors, and the evaluation has been overseen by Gisela Storz as the Senior Editor. The following individual involved in review of your submission has agreed to reveal their identity: Joe Bondy-Denomy (Reviewer #2).

The reviewers have discussed the reviews with one another, and the Reviewing Editor has drafted this decision to help you prepare a revised submission.

We would like to draw your attention to changes in our revision policy that we have made in response to COVID-19 (https://elifesciences.org/articles/57162). Specifically, we are asking editors to accept manuscripts that they judge can stand as *eLife* papers without additional data, even if they feel that additional experiments would make the manuscript stronger. While the reviewers have identified additional experiments that would improve this work, and you are welcome to add these to the revised manuscript if conditions allow, the only required experiments are computational in nature.

Summary:

The authors construct a series of genetic mutants and use molecular methods to show that CRISPR transcription is terminated by Rho and that the Nus complex blocks Rho-dependent termination, resulting in longer CRISPR transcripts. The paper is well written, the results are convincing, and the work is of interest of a broad audience. However, it is not clear why the competing activities of Rho and Nus are important for regulation of these systems. When and where is the regulation of Rho and Nus relevant to the biological control of CRISPR expression? Are expression levels of Rho and Nus controlled in response to phage infection or plasmid conjugation?

Essential revisions:

Please include data for the H-NS knockout or explain why this is not included. Explain where the natural transcriptional start sites are located relative to the engineered promoter. Leader sequences are typically AT-rich and contain the promotor. What is the relevance of a GC-rich "Rut" if it is upstream of the natural promoter?

The authors detect SuhB occupancy in the CRISPR-II *boxA* region of S. Typhimurium (Figure 1), and from that observation conclude that Nus factors are also involved. SuhB is a relatively new addition to the Nus complex and only recently found to play a role in rRNA expression (Singh et al., 2016). It is unclear whether SuhB is always associated with Nus antitermination complexes, or if it might work alone, or with other factors at different promoters. Given this uncertainty and caveats of the Nus deletion/mutation experiments presented here, it is essential that the authors either perform additional experiments to test for relevant Nus factor(s) at the putative *boxA* sequences of both CRISPR arrays in S. Typhimurium, or revise the text to be more clear about the role of SuhB and the inferred role of Nus.

The authors show the high conservation of NusB in bacteria (Figure 7—figure supplement 1) to support the notion that Nus-mediated antitermination is a general mechanism employed in diverse CRISPR loci. However, it was SuhB which was detected at the CRISPR-II *boxA* sequence. The phylogenetic analysis should be performed on SuhB. Phylogenetic comparisons of CRISPRs, BoxA, SuhB and NusB may be necessary to speculate about the widespread distribution of this mechanism.

Figure 2B: Given the large difference in BCM treatment for the long construct, which has an ~10 fold impact over the BoxA mutation (which also has a 10-fold impact), this strong effect of the drug might necessitate a control transcript/fusion to assess the uptick in *lacZ* from any locus? Or is this BCM effect specific to the CRISPR array and therefore there is a second (or third or fourth) equally potent/important site that the authors have not identified?

Clarify why some experiments were only performed with CRISPR-II (Figures 1 and 2), while some were only performed with CRISPR-I (Figure 3 and Figure 1—figure supplement 1). Observations made about one locus were assumed to apply to the other. Include data for SuhB/Nus occupancy at *boxA* sequences for both CRISPR loci, and the effects of bicyclomycin on expression for each loci or explain why this has been omitted.

The *boxA*-dependent stimulation of promoter-distal spacer activity is assumed to be through a mechanism of antitermination. However, the data might equally support the possibility that SuhB facilitates/stimulates crRNA maturation rather than promoting antitermination, much like what was discovered at the rRNA operon (Singh et al., 2016). To distinguish between the two possibilities, it is necessary to check for RNAP occupancy at promoter -proximal vs -distal spacers or test the efficiency of CRISPR RNA processing with and without SuhB. Alternatively, temper the conclusion that the mechanism is antitermination SuhB dependent antitermination.

Clarify the statistical methods used in the ChIP-seq experiments. For example, in Figure 4A, it appears as though the purple data points indeed cluster away from the orange, but in Figure 4—figure supplement 2A, it is less clear which of the blue data points cluster away from the orange data points in a statistically significant manner.

Throughout the manuscript, it is implied that the proposed antitermination mechanism occurs in all CRISPR-Cas types, while experimental data was collected for only two Type I systems. It is important to explicitly state which CRISPR Type(s) were found to harbor *boxA* sequences (Figure 7A) to support the possibility that a general mechanism has been discovered and to clarify how this conclusion relates to the work presented by Lin et al., 2019.

Figure 7 was generated while working under the assumption that CRISPR arrays appear downstream of *cas2*. While this may be true for some CRISPR-Cas systems, this approach excludes many systems with slightly different genomic architectures. For a more unbiased approach, search for *boxA* sequences directly upstream of the first repeat in a CRISPR array. This approach is anticipated to provide more reliable evidence to support the general prevalence of *boxA* sequences upstream of CRISPR arrays.

The authors speculate that "Rho termination acts as a selective pressure to limit adaptation in species that lack an antitermination mechanism". However, the possible role of Rho in limiting adaption seems indirect at best. If Rho-dependent termination limits the number of different spacers that can be expressed from a single locus then this will limit selective pressures that maintain "older spacers", but the advantage this afford the host is unclear. Furthermore, role of Nus would be expected to have an opposing impact on CRISPR length, so it is unclear how this explains an abundance of short CRISPRs and the authors do not clarify how these observations fit with the numerous genomes that do have long CRISPRs.

Results first paragraph, specify what CRISPR-Cas Type and subtype you are working with.

[Editors' note: further revisions were suggested prior to acceptance, as described below.]

Thank you for resubmitting your work entitled "Transcription Termination and Antitermination of Bacterial CRISPR Arrays" for further consideration by *eLife*. Your revised article has been evaluated by Gisela Storz (Senior Editor) and Blake Wiedenheft (Reviewing Editor).

We have reviewed the rebuttal and the revised manuscript. The revision sufficiently addresses the reviewer's concerns, with one important exception.

The authors provide convincing experimental evidence for the competing roles of Rho and Nus in CRISPR transcription, but I am still not convinced by the arguments about the how these factors impact CRISPR length.

One of the reviewers raised this concern during the review. They pointed to the following statement: "Rho termination acts as a selective pressure to limit adaptation in species that lack an antitermination mechanism". However, as the reviewer pointed out, "the possible role of Rho in limiting adaption seems indirect at best." In the rebuttal, the authors address the comment by stating that "Our data are insufficient to conclude that the presence of Rho causes many CRISPR arrays to be short, but we think this is likely in cases where there isn't an antitermination mechanism, and hence we discuss this, clearly framed as speculation." I agree that it is appropriate to speculate in the Discussion, but the third sentence of the Abstract states "We show that Rho termination functionally limits the length of bacterial CRISPR arrays". Data presented by the authors, clearly demonstrates that Rho limits the length of CRISPR transcripts and Nus antagonizes Rho-dependent termination, but as the reviewer points out, "the possible role of Rho in limiting adaption (i.e., length of the CRISPR locus) seems indirect at best".

Please clarify statements connecting the role of Rho to CRISPR length. Speculation should be omitted form the Abstract. In addition, please clarify the following statement "type II-C CRISPR-Cas systems have their CRISPR arrays oriented opposite to *cas3*". I suspect that the context of this statement is important, but I have read this several times and it still seems to me like the authors are suggestion that type-II systems have a *cas3*. Please clarify.

---

## [Author Response]

Summary:The authors construct a series of genetic mutants and use molecular methods to show that CRISPR transcription is terminated by Rho and that the Nus complex blocks Rho-dependent termination, resulting in longer CRISPR transcripts. The paper is well written, the results are convincing, and the work is of interest of a broad audience. However, it is not clear why the competing activities of Rho and Nus are important for regulation of these systems. When and where is the regulation of Rho and Nus relevant to the biological control of CRISPR expression. Are expression levels of Rho and Nus controlled in response to phage infection or plasmid conjugation?

We have expanded the Discussion to speculate on the significance of Rho termination and antitermination for crRNA expression. We believe that Rho termination, a highly conserved and often essential activity in bacteria, inevitably poses a barrier to the expression of longer CRISPR arrays, necessitating an antitermination mechanism. Whether antitermination is subject to regulation in response to environmental cues such as invading phage/plasmids is unknown, but we include this possibility as speculation.

Essential revisions:Please include data for the H-NS knockout or explain why this is not included. Explain where the natural transcriptional start sites are located relative to the engineered promoter. Leader sequences are typically AT-rich and contain the promotor. What is the relevance of a GC-rich "Rut" if it is upstream of the natural promoter?

The *cas* genes and both CRISPR arrays in *Salmonella* are transcriptionally silenced by H-NS; there is no known in vitro growth condition that induces their expression. Hence, we are forced to rely on mutant strains to study the *Salmonella* CRISPR-Cas system. We chose to introduce promoters in the chromosome. For the CRISPR-I array, we inserted the promoter upstream of *cas8e*, and for the CRISPR-II array we inserted the promoter immediately downstream of *queE*. Thus, these promoters mimic transcription read-through from the *cas* operon and *queE* mRNAs, respectively. Given the requirement that a Rut be located in untranslated RNA, we presume that the Rut for CRISPR-I is downstream of *cas2*, and the Rut for CRISPR-I is downstream of *queE*. We note that a previous study of the *Escherichia coli* CRISPR-Cas system identified a promoter within the leader of the CRISPR-I array. However, our ChIP-qPCR data for RNA polymerase (Figure 1—figure supplement 1) clearly indicate that the majority of CRISPR array transcription in our engineered strain of *Salmonella* comes from read-through of the *cas* gene mRNA. Hence, in situations where the *cas* genes are transcribed, we would expect considerable read-through into the CRISPR array. We have expanded the description of the first supplementary figure to highlight the significance of this result.

Deleting *hns* would presumably induce expression of both CRISPR arrays, but there are several reasons to avoid this strategy. First, *hns* is essential in *Salmonella*, so deleting *hns* would require a secondary mutation. Second, even with a secondary mutation, deleting *hns* causes a large growth defect, and likely has pleiotropic effects. Third, we have shown previously that H-NS silences many cryptic promoters within genes in *E. coli*, so deleting *hns* might result in transcription from promoters within the *cas* genes.

The authors detect SuhB occupancy in the CRISPR-II boxA region of S. Typhimurium (Figure 1), and from that observation conclude that Nus factors are also involved. SuhB is a relatively new addition to the Nus complex and only recently found to play a role in rRNA expression (Singh et al., 2016). It is unclear whether SuhB is always associated with Nus antitermination complexes, or if it might work alone, or with other factors at different promoters. Given this uncertainty and caveats of the Nus deletion/mutation experiments presented here, it is essential that the authors either perform additional experiments to test for relevant Nus factor(s) at the putative boxA sequences of both CRISPR arrays in S. Typhimurium, or revise the text to be more clear about the role of SuhB and the inferred role of Nus.

We and others have shown that SuhB is an integral part of the Nus factor complex. SuhB is recruited to elongating RNAP complexes in a BoxA-dependent manner, and extensive genetic data are consistent with an essential role within the complex. We note that three recent structural studies (Dudenhoeffer et al., 2019, Huang et al., 2019 and Huang et al., 2020) show that SuhB is required for assembly and activity of the Nus factor complex. We have expanded our description of the Nus factor complex, and specifically the role of SuhB, to clarify this point.

The authors show the high conservation of NusB in bacteria (Figure 7—figure supplement 1) to support the notion that Nus-mediated antitermination is a general mechanism employed in diverse CRISPR loci. However, it was SuhB which was detected at the CRISPR-II boxA sequence. The phylogenetic analysis should be performed on SuhB. Phylogenetic comparisons of CRISPRs, BoxA, SuhB and NusB may be necessary to speculate about the widespread distribution of this mechanism.

All the proteins in the Nus factor complex, with the exception of NusB, are known to have functions outside the complex. SuhB is an inositol monophosphatase. A phylogenetic comparison of *suhB* could be misleading, since SuhB could be conserved due to its function as an inositol monophosphatase. We now explain in the text why we chose to only look at conservation of *nusB*.

Figure 2B: Given the large difference in BCM treatment for the long construct, which has an ~10 fold impact over the BoxA mutation (which also has a 10-fold impact), this strong effect of the drug might necessitate a control transcript/fusion to assess the uptick in lacZ from any locus? Or is this BCM effect specific to the CRISPR array and therefore there is a second (or third or fourth) equally potent/important site that the authors have not identified?

We agree with this concern and we have assayed an additional *lacZ* fusion where the CRISPR array sequence (including the upstream region) has been replaced with an intrinsic terminator from *Escherichia coli*. We have previously shown that this terminator substantially reduces, but does not abolish, expression of the reporter gene. We chose to include an intrinsic terminator because it sensitizes the reporter to increases in expression. The control reporter showed no change in expression upon addition of BCM. These data are now included in Figure 2.

Clarify why some experiments were only performed with CRISPR-II (Figures 1 and 2), while some were only performed with CRISPR-I (Figure 3 and Figure 1—figure supplement 1). Observations made about one locus were assumed to apply to the other. Include data for SuhB/Nus occupancy at boxA sequences for both CRISPR loci, and the effects of bicyclomycin on expression for each loci or explain why this has been omitted.

We only looked at SuhB occupancy at one locus for the sake of simplicity. The strains required for these experiments are not trivial to make – they require multiple rounds of recombineering – and we reasoned that the result from one array could likely be extrapolated to the other, especially in light of the multiple, independent lines of evidence linking the Nus factor complex to both CRISPR arrays.

The boxA-dependent stimulation of promoter-distal spacer activity is assumed to be through a mechanism of antitermination. However, the data might equally support the possibility that SuhB facilitates/stimulates crRNA maturation rather than promoting antitermination, much like what was discovered at the rRNA operon (Singh et al., 2016). To distinguish between the two possibilities, it is necessary to check for RNAP occupancy at promoter -proximal vs -distal spacers or test the efficiency of CRISPR RNA processing with and without SuhB. Alternatively, temper the conclusion that the mechanism is antitermination SuhB dependent antitermination.

We looked at RNAP occupancy using ChIP-qPCR (data not in the manuscript), but the absolute occupancy numbers are too low to confidently conclude anything from these data. Hence, we chose to use reporter assays, which are more sensitive. We believe the reporter assays make a strong case that it is the antitermination function that is required for activity of the leader-distal spacers. Two additional lines of evidence support an antitermination function for *boxA* sequences. First, the patterns of spacer usage in the *boxA* mutants of both *Salmonella* and *Vibrio* are consistent with premature transcription termination. Second, the apparent termination within the CRISPR array of the *SalmonellaboxA* mutants is reversed by bicyclomycin, a Rho inhibitor. Nonetheless, we have added some text to leave open the possibility that the Nus factor complex might affect crRNA maturation either instead of, or in addition to antitermination. This is particularly relevant in light of an observation we made when revisiting the conservation of *boxA* sequences upstream of CRISPR arrays. Many cyanobacterial species appear to have a *boxA* upstream of their CRISPR arrays, but these species also lack *rho*, a point that had previously escaped our attention. We were concerned that the sequences upstream of cyanobacterial CRISPR arrays might not be genuine *boxA* sequences, especially since these sequences differ at key positions from the consensus *boxA* sequence derived from *E. coli*. To more thoroughly assess whether the cyanobacterial sequences are genuine *boxA* sequences, we searched for conserved sequences upstream of ribosomal RNA loci in the same set of species, based on the assumption that each ribosomal RNA upstream region would likely contain one *boxA* sequence. The most enriched sequence from cyanobacterial ribosomal RNA upstream regions is a very close match to the putative *boxA* sequences upstream of the cyanobacterial CRISPR arrays, and there are no enriched sequences that more closely resemble the *E. coliboxA* consensus; hence, we are confident that the cyanobacterial *boxA* sequences upstream of CRISPR arrays are genuine. These data are included in a new supplementary figure. Given that cyanobacterial species lack *rho*, we conclude that the BoxA elements in cyanobacterial CRISPR array RNAs have a function that is distinct from antagonizing Rho. We propose two possible functions. First, there may be an unknown transcription termination mechanism in cyanobacteria that the Nus factor complex can inhibit. Second, the Nus factor complex may be required to loop the RNA between the BoxA sequence and the transcribing RNA polymerase, perhaps to promote crRNA processing. This latter possibility could also apply in species that have *rho*.

Clarify the statistical methods used in the ChIP-seq experiments. For example, in Figure 4A, it appears as though the purple data points indeed cluster away from the orange, but in Figure 4—figure supplement 2A, it is less clear which of the blue data points cluster away from the orange data points in a statistically significant manner.

We considered using a statistical test to determine which datapoints in Figures 4, 5, Figure 3—figure supplement 1 and Figure 4—figure supplement 2 have significantly lower Cas5 occupancy than expected by chance. However, this is not a straightforward analysis, largely because we did not define a null hypothesis before doing the experiment. We think the magnitude of the effects observed, and the trends as they relate to spacer position within the two arrays, are more important than *p*-values, and we hope the data speak for themselves in that regard. We note that the ratios of Cas5 binding in *boxA* mutant vs wild-type strains are shown in Figures 4B and Figure 4—figure supplement 2B. We do not show these ratios in Figure 5 because there is nothing to normalize to. Lastly, we have reworded three instances where we previously described differences as “significant”.

Throughout the manuscript, it is implied that the proposed antitermination mechanism occurs in all CRISPR-Cas types, while experimental data was collected for only two Type I systems. It is important to explicitly state which CRISPR Type(s) were found to harbor boxA sequences (Figure 7A) to support the possibility that a general mechanism has been discovered and to clarify how this conclusion relates to the work presented by Lin et al., 2019.

This is a very interesting point, and one we had not considered. We have modified Supplementary file 4 to show the different types of CRISPR-Cas system associated with each species listed. Of the 187 species tested, only 17 lack a type I CRISPR-Cas system (many have multiple types). 15 of these 17 species have only a type II-C system, and none of these 15 species have an identified *boxA* sequence between *cas2* and their CRISPR arrays. Thus, their appears to be a mutually exclusive relationship between type II-C systems and the presence of *boxA* sequences between *cas2* and CRISPR arrays. Beyond this, it is difficult to conclude much. There are two species with only a type III system that have a putative *boxA*, suggesting that *boxA*-mediated antitermination is not exclusive to type I systems.

We wondered why type II-C systems might lack *boxA* sequences. For the two well-characterized type II-C systems, the CRISPR array is adjacent to *cas2*, but is unusual in that it is transcribed in the opposite orientation to *cas2*. Hence, the sequences we used to search for *boxA* sequences may be from the wrong side of the type II-C CRISPR arrays. To test this possibility, we repeated the search for enriched sequence motifs using 300 nt sequences from the other side of the 15 type II-C arrays (sequences from other 172 species were unchanged). We did not identify a *boxA* sequence adjacent to any of the 15 type II-C arrays. Previous work on type II-C systems has shown that some type II-C repeats contain a promoter, such that each spacer has its own promoter immediately upstream. Not many type II-C systems have been studied, but it has been suggested that this phenomenon is true for most type II-C repeats. We suspect that this provides an alternative way to avoid Rho termination. We now raise this possibility in the Discussion.

While there are clearly parallels between our work and that of Lin *et al.*, we have identified a processive antitermination mechanism (i.e. a mechanism that modifies the RNA polymerase to make it resistant to Rho throughout the transcription unit) that applies to entire CRISPR arrays and is phylogenetically widespread, whereas Lin *et al.* identified a targeted antitermination mechanism (i.e. a mechanism that prevents one instance of Rho loading) that applies to leader sequences in a narrow range of species. A processive antitermination mechanism rather than a targeted antitermination mechanism is important because the sequences of newly acquired spacers cannot be anticipated.

Figure 7 was generated while working under the assumption that CRISPR arrays appear downstream of cas2. While this may be true for some CRISPR-Cas systems, this approach excludes many systems with slightly different genomic architectures. For a more unbiased approach, search for boxA sequences directly upstream of the first repeat in a CRISPR array. This approach is anticipated to provide more reliable evidence to support the general prevalence of boxA sequences upstream of CRISPR arrays.

We agree that this would be an interesting analysis, but it is technically a lot more challenging than the analysis focused on *cas2*. For example, we would need to determine the orientation of every CRISPR array, and we would need to define an upstream window to search within. We actually tried to do this analysis, but struggled to find a way to extract sequences upstream of all CRISPR arrays in batch, and to get the correct orientation. Aligning the CRISPR array relative to *cas2* likely identifies the correct orientation in most cases, with the exception of type II-C systems described above. Moreover, *cas2* provides a natural upstream barrier. We believe that the existing analysis makes a strong enough case that BoxA-mediated antitermination is phylogenetically widespread.

The authors speculate that "Rho termination acts as a selective pressure to limit adaptation in species that lack an antitermination mechanism". However, the possible role of Rho in limiting adaption seems indirect at best. If Rho-dependent termination limits the number of different spacers that can be expressed from a single locus then this will limit selective pressures that maintain "older spacers", but the advantage this afford the host is unclear. Furthermore, role of Nus would be expected to have an opposing impact on CRISPR length, so it is unclear how this explains an abundance of short CRISPRs and the authors do not clarify how these observations fit with the numerous genomes that do have long CRISPRs.

Our data are insufficient to conclude that the presence of Rho causes many CRISPR arrays to be short, but we think this is likely in cases where there isn’t an antitermination mechanism, and hence we discuss this, clearly framed as speculation. Of note, we show that in *Salmonella*, mutating the *boxA* can impact spacers as early as spacer 9 (relative to the leader), and potentially further upstream than that (we have no data for spacers 3-8 in the CRISPR-II array). We also cannot say with any degree of certainty that other antitermination mechanisms exist, but we suspect they do, and that would explain how some species with *rho* have long CRISPR arrays. We think it is possible that many species lack any antitermination mechanism for their CRISPR arrays, which would be consistent with them having short arrays. Again, we are careful to frame this as speculation.

Results first paragraph, specify what CRISPR-Cas Type and subtype you are working with.

Fixed.

[Editors' note: further revisions were suggested prior to acceptance, as described below.]

The authors provide convincing experimental evidence for the competing roles of Rho and Nus in CRISPR transcription, but I am still not convinced by the arguments about the how these factors impact CRISPR length.One of the reviewers raised this concern during the review. They pointed to the following statement: "Rho termination acts as a selective pressure to limit adaptation in species that lack an antitermination mechanism". However, as the reviewer pointed out, "the possible role of Rho in limiting adaption seems indirect at best." In the rebuttal, the authors address the comment by stating that "Our data are insufficient to conclude that the presence of Rho causes many CRISPR arrays to be short, but we think this is likely in cases where there isn't an antitermination mechanism, and hence we discuss this, clearly framed as speculation." I agree that it is appropriate to speculate in the Discussion, but the third sentence of the Abstract states "We show that Rho termination functionally limits the length of bacterial CRISPR arrays". Data presented by the authors, clearly demonstrates that Rho limits the length of CRISPR transcripts and Nus antagonizes Rho-dependent termination, but as the reviewer points out, "the possible role of Rho in limiting adaption (i.e., length of the CRISPR locus) seems indirect at best".Please clarify statements connecting the role of Rho to CRISPR length. Speculation should be omitted form the Abstract. In addition, please clarify the following statement "type II-C CRISPR-Cas systems have their CRISPR arrays oriented opposite to cas3". I suspect that the context of this statement is important, but I have read this several times and it still seems to me like the authors are suggestion that type-II systems have a cas3. Please clarify.

We have modified the Abstract, which now states “We show that Rho can prematurely terminate transcription of bacterial CRISPR arrays” rather than “We show that Rho termination functionally limits the length of bacterial CRISPR arrays”. We have also added “and antitermination” to the last sentence of the Abstract.

The statement “type II-C CRISPR-Cas systems have their CRISPR arrays oriented opposite to *cas3*” has a typo – it should be “*cas2*” rather than “*cas3*”; we have fixed this error, and added the clarifying text: “when extracting sequences adjacent to CRISPR arrays for sequence analysis (Figure 7A), we assumed that *cas2* and CRISPR arrays are oriented co-directionally”. In other words, even if there had been *boxA* sequences upstream of the CRISPR arrays for species with a type II-C system, we were probably looking on the wrong site of the array in our initial analysis.